# Immunogenicity and safety of a recombinant adenovirus type-5 COVID-19 vaccine in adults: Data from a randomised, double-blind, placebo-controlled, single-dose, phase 3 trial in Russia

Dmitry Lioznov[1,2], Irina Amosova[1], Savely A. Sheetikov[3], Ksenia V. Zornikova[3], Yana Serdyuk[3], Grigory A. Efimov[3], Mikhail Tsyferov[4], Mikhail Khmelevskii[4], Andrei Afanasiev[4], Nadezhda Khomyakova[4], Dmitry Zubkov[4], Anton Tikhonov[4], Tao Zhu[5], Luis Barreto[5], Vitalina Dzutseva[4,6]*

1 Smorodintsev Research Institute of Influenza, St Petersburg, Russia, 2 Department of Infectious Diseases and Epidemiology, First Pavlov State Medical University, St. Petersburg, Russia, 3 National Research Center for Hematology, Moscow, Russia, 4 NPO Petrovax Pharm LLC, Moscow, Russia, 5 CanSino Biologics Inc., Tianjin, China, 6 Novosibirsk State University, Medical School, Novosibirsk, Russia

* DzutsevaVV@petrovax.ru

## Abstract

### Background

To determine the immunogenicity, efficacy, reactogenicity, and safety of a single dose of recombinant adenovirus type-5 vectored COVID-19 vaccine (Ad5-nCoV, $5 \times 10^{10}$ viral particles per 0.5 mL dose), we conducted a single-dose, randomised, double-blind, placebo-controlled, parallel group (3:1 Ad5-nCoV:placebo), phase 3 trial (Prometheus).

### Methods

From 11-September-2020 to 05-May-2021, across six sites in the Russian Federation, 496 participants were injected with either placebo or Ad5-nCoV expressing the full-length spike (S) protein from the severe acute respiratory syndrome coronavirus 2 (SARS-CoV-2).

### Results

Seroconversion (the primary endpoint) rates of 78.5% (95% CI: 73.9; 82.6) against receptor binding domain (RBD), 90.6% (95% CI: 87.2; 93.4) against S protein and 59.0% (95% CI: 53.3; 64.6) seroconversion of neutralising antibodies against SARS-CoV-2 at 28 days post-vaccination were observed. Geometric mean titres (GMTs) were also elevated for antibodies against the RBD (405 [95% CI: 366; 449]) and S protein (677 [95% CI: 608; 753]) compared to the GMT of neutralising antibodies against SARS-CoV-2 (16.7 [95% CI: 15.3; 18.3]). Using an IFN-γ ELISpot assay after stimulating the cells with recombinant S protein ectodomain we showed that the Ad5-nCoV vaccine induced the most robust cellular immune response on Days 14 and 28. Up to Day 28, the primary and all secondary

**Data Availability Statement:** All relevant data are within the manuscript and its Supporting Information files.

**Funding:** The study was designed, funded, and managed by NPO Petrovax Pharm LLC (Moscow, Russian Federation). NPO Petrovax Pharm LLC in partnership with CanSino Biologics, Inc. (Tianjin, China) are funding and managing the clinical development of the Ad5-nCoV vaccine in the Russian Federation. MT, MK, DZ, AA, NK, AT and VD are employees of NPO Petrovax Pharm LLC. TZ and LB are employees of CanSino Biologics, Inc. IA, SS, KZ and YS have received funding from NPO Petrovax Pharm LLC for consultation services. DL and GE have received personal fees from NPO Petrovax Pharm LLC for consultation services. The funders had no additional role in study design, data collection and analysis, decision to publish, or preparation of the manuscript. The specific roles of these authors are articulated in the 'author contributions' section.

**Competing interests:** MT, MK, DZ, AA, NK, AT and VD are employees of NPO Petrovax Pharm LLC. TZ and LB are employees of CanSino Biologics, Inc. IA, SS, KZ and YS have received funding from NPO Petrovax Pharm LLC for consultation services. DL and GE have received personal fees from NPO Petrovax Pharm LLC for consultation services. NPO Petrovax Pharm LLC in partnership with CanSino Biologics, Inc. are funding and managing the clinical development of the Ad5-nCoV vaccine in the Russian Federation. This does not alter our adherence to PLOS ONE policies on sharing data and materials.

endpoints of the Ad5-nCoV vaccine were statistically significant compared with the placebo (<0.001). Systemic reactions were reported in 113 of 496 (22.8%) participants (Ad5-nCoV, 26.9%; Placebo, 10.5%), and local reactions were reported in 108 (21.8%) participants (Ad5-nCoV, 28.5%; Placebo, 1.6%). These were generally mild and resolved within 7 days after vaccination. Of the six serious adverse events reported, none of the events were vaccine related. There were no deaths or premature withdrawals.

## Conclusion

A single-dose of Ad5-nCoV vaccine induced a marked specific humoral and cellular immune response with a favourable safety profile.

## Trial registration

**Trial registration:** ClinicalTrials.gov: NCT04540419.

## Introduction

The ongoing COVID-19 pandemic caused by severe acute respiratory syndrome coronavirus 2 (SARS-CoV-2) has resulted in morbidity and mortality unseen since the Spanish flu outbreak more than a century ago [1]. Quarantine measures, a traditional public health method of infection control, have been only partially effective in the face of this highly transmissible virus [2]. Vaccination has the potential to reduce disease severity and transmission but requires expedited development and global administration. In response to this enormous task, many potential vaccine candidates are in development, with several now approved for full or limited use being actively administered [3–6]. A range of technologies have been used in the development of SARS-CoV-2 vaccines, including replicating or non-replicating viral vectors, inactivated viruses and mRNA, DNA or autologous cell-based vaccines. It is not currently known which approach provides the most effective immunity for different recipient risk groups with an acceptable safety profile, all at acceptable levels of cost with ease of manufacturing and distribution [7].

A candidate COVID-19 vaccine that initially showed a capacity to induce a significant antibody and cellular immune response is the adenovirus type 5 (Ad5)-nCoV vaccine developed by CanSino Biologics Inc., Tianjin, and the Beijing Institute of Biotechnology, People's Republic of China. It consists of a replication-defective Ad5 vector expressing the SARS-CoV-2 spike (S) protein, including its receptor-binding domain (RBD). This protein serves as the main antigen in SARS-CoV-2 vaccines [8]. The Ad5-nCoV vaccine was one of the first to enter phase 1 and 2 clinical trials in China, 2 months after the identification of the virus genotype and the results showed the vaccine to be safe and immunogenic after a single dose [9,10]. Its potential advantages are single-dose immunization and its proven technology, both of which were used previously for the Ebola vaccine (Ad5-EBOV), and its stability that permits it to be stored in a standard refrigerator at 2–8˚C, enabling the ease of worldwide distribution.

A collaborative development project between CanSino Biologics and the Russian pharmaceutical company NPO Petrovax Pharm LLC provided the basis for a new phase 3 study, Prometheus Rus, which is the clinical protocol code that was approved by a local authority. This multicentre, randomised, double-blind, placebo-controlled clinical trial examines the immunogenicity, reactogenicity, efficacy and safety of the Ad5-nCoV COVID-19 vaccine compared

with placebo, in a mostly white, Russian population. The study population was recruited at centres in the western part of the Russian federation (Moscow, St. Petersburg and Yaroslavl) and therefore provides the first clinical data available for Ad5-nCoV in a white European population; previously published data were based on the phase 1 and 2 clinical trials conducted in China. We present the final analysis of the 496 participants in this phase 3 trial.

## Materials and methods

The protocol for this trial and supporting CONSORT checklist are available as supporting information; see S1 Protocol and S1 Checklist, respectively. The Ad5-nCoV vaccine and placebo drug products were produced by CanSino Biologics Inc [8]. Secondary packaging and quality control were performed by NPO Petrovax.

### Ethical conduct of the study

The study proceeded in accordance with the principles of the Declaration of Helsinki and Good Clinical Practice. The trial protocol was reviewed and approved by the Independent Ethics Committees of the involved sites and the Ethics Council of the Ministry of Health of the Russian Federation. The study design and methodology have been developed in line with the FDA [11], EMA [12] and EAEU [13] guidelines, as well as the regulatory documents of the Russian Federation [14]. The trial is registered with ClinicalTrials.gov, NCT04540419 [15].

To be included, participants needed to be able to understand the content of the informed consent documents and be willing to sign the informed consent form. Written informed consent was obtained from each participant before eligibility screening.

### Study design and participants

Prometheus is a multicentre, randomised, double-blind, placebo-controlled, clinical trial being conducted in six centres in the Russian Federation. The study seeks to evaluate the immunogenicity, efficacy, reactogenicity and safety of a single dose of the Ad5-nCoV COVID-19 vaccine compared with placebo in adults up to 6 months after vaccination. Competitive recruitment of the planned sample size of 500 eligible participants in six centres was completed in November 2020 and follow-up observations were completed on 05-May-2021. A planned interim analysis of data collected in 200 participants up to 28 days after a single injection of Ad5-nCoV or placebo was completed on 21-December-2020. Herein, we present the results of final analysis of data collected in 496 participants up to 6 months after a single injection of Ad5-nCoV or placebo.

Participation was sought through online recruitment advertising and patient databases of the trial sites. All participants underwent detailed screening 1–10 days before vaccination with Ad5-nCoV or placebo (Day 0). Screening included the detection of SARS-CoV-2 RNA using real-time polymerase chain reaction (PCR) via a swab, and SARS-CoV-2 immunoglobulin M (IgM) and immunoglobulin G (IgG) antibody testing to ensure negative results, as well as testing for human immunodeficiency virus (HIV), syphilis, hepatitis B and hepatitis C viruses via blood serum. A detailed medical history for each participant was taken and records included if the participant experienced any COVID-19 symptoms and if the participant was in close contact with people suspected or proven to have SARS-CoV-2 infection. Participants underwent physical examination (including a neurological examination, vital signs, and body temperature), haematological, biochemical and coagulation testing, urine analysis, electrocardiogram, and when applicable, pregnancy testing as pregnancy (and breastfeeding) was an exclusion criterion.

Men and women aged 18–85 years with a body mass index (BMI) between 18.5 and 30.0 kg/m$^2$ were selected to participate if they had no indication of a current or previous SARS-CoV-2 infection (e.g., respiratory infection in last 14 days, axillary temperature ≥37.0˚C) or close contact with a suspected or confirmed case of SARS-CoV-2 infection. Participants were considered to be eligible if they were in general good health as established by medical history and screening. Those with a range of chronic illnesses, including mental disorders, a history of allergies, recipients of concurrent medication, those with addictions, and those for whom there were concerns over adherence to study protocol were also excluded.

## Randomisation and study populations

The investigational vaccine, Ad5-nCoV, and the placebo were provided by NPO Petrovax Pharm LLC (Moscow, Russia). Both vaccine and placebo were developed by CanSino Biologics Inc. (Tianjin, China) and the Beijing Institute of Biotechnology (Beijing, China). The vaccine was administered with the optimal dose of $5 \times 10^{10}$ viral particles per 0.5 mL dose, as determined in a previous study [9,10]; placebo contained vaccine excipients only. The appearance of Ad5-nCoV and placebo syringes and packaging was identical.

Eligible participants were randomly allocated to the Ad5-nCoV group or the Placebo group, in a 3:1 ratio, by an independent statistician using a validated system including a pseudorandom number generator with a seed value; allocation used block randomisation and stratification by study site. Neither the investigators nor participants were aware of the group assignment. Investigators were trained to use the centralised interactive web response system that was used for randomisation. Randomisation codes were kept by authorised personnel from the responsible contracted organisation.

The Safety Analysis Set included all randomised participants who received a dose of the vaccine and was used to provide the disposition of study participants. Results for immunogenicity analyses are presented for the full analysis set ([FAS] for immunogenicity analysis), which included all eligible participants who received a dose of vaccine and provided at least one immunogenicity assessment result. The study design (S1 Protocol) also included the per-protocol sets ([PPS] for immunogenicity analysis and PPS efficacy analysis), which included members of the FAS for immunogenicity analysis and Safety Analysis Set, respectively, with no significant protocol deviations and who did not develop COVID-19 within the first 14 days post-vaccination. The population characteristics and results were similar between the FAS for immunogenicity analysis and both types of PPS (immunogenicity and efficacy analysis); therefore, only the FAS for immunogenicity analysis population was used to evaluate the immunogenicity endpoints in this study. Cellular immunity results were analysed in a subset of participants from the FAS for immunogenicity analysis population that attended the Moscow clinic site (n = 69). Details describing the sample sizes are provided in S1 Methods.

## Procedures

A single dose (0.5 mL) of Ad5-nCoV or placebo was administered by intramuscular (IM) injection to the upper arm on Day 0. Participants were requested to remain at the site for 2 h after vaccination for study staff to monitor for any systemic or local reactions to vaccination. Following the administration of the vaccine or placebo, participants attended clinic visits on Day 2, Day 7, Day 14, Day 28 and after Month 6. Between Day 28 and Month 6 there were phone calls at Months 2, 3, 4 and 5.

Determination of serum antibodies against the S protein and RBD of SARS-CoV-2 and the presence of neutralising antibodies against SARS-CoV-2 (NAbs) were conducted on Day 0, Day 14, Day 28 and after Month 6; neutralising antibodies against Ad5 were assessed on Day

0, Day 28 and after Month 6. Immunoglobulin G antibodies to the S protein and RBD were determined using indirect enzyme-linked immunosorbent assay kits that involved incubation of serially diluted serum samples with the recombinant antigen (either RBD [SARS-CoV-2-IgG-EIA; XEMA Co. Ltd] or S protein [SARS-CoV-2-IgG-EIA-BEST; Vector-Best]) immobilised to the surface of a 96-well plate. The assays were performed according to the manufacturer's instructions. Horseradish-peroxidase conjugated mouse monoclonal anti-human IgG antibody was used to detect antibodies, visualised with tetramethylbenzidine substrate solution. The detection limit for antibodies against the S protein and RBD of SARS-CoV-2 was 1:100.

Anti-SARS-CoV-2 neutralising antibodies were determined with a microneutralisation assay in which Vero cell (#L-81, American Type Culture Collection) monolayers were incubated in 96-well plates with 2-fold serial dilutions (1:10 to 1:1280) of participant serum. Recently thawed and diluted SARS-CoV-2 virus ([GISAID HCoV-19/st_petersburg-3524S/2020] was obtained from the clinical material collection at Smorodintsev Research Institute of Influenza, St. Petersburg, Russia) was added to the wells, and plates were incubated for 1 h at $37\pm0.5°C$ in a humidified incubator. Medium was removed from wells, replaced with a mixture of the most highly diluted (1:1280) serum and virus, and plates were incubated at $37\pm0.5°C$ and 5% $CO_2$ for 4 days. Anti-Ad5 neutralising antibodies were also determined using microneutralisation assay; A549 cell monolayers were incubated in 96-well plates with serial dilutions (1:10 to 1:1280) of participant serum and working dilutions of Ad5 (Adenovir; Smorodintsev Research Institute of Influenza). Plates were incubated for 2 h at $36\pm0.5°C$ and 5% $CO_2$. Medium was removed from wells, replaced with a mixture of diluted sera and working virus dilution, and plates were incubated at $36\pm0.5°C$ and 5% $CO_2$ for 72 h. For both microneutralisation assays, results were assessed by visual inspection of cytopathic effects. Serum titres were determined as the maximum dilution at which complete inhibition of viral propagation was detected as the result of interaction between virus and specific antibodies; the detection limit was 1:10. Antibody titres undetectable in serum were assigned values of half of the detection limits for calculation.

Sequence encoding ΔFurin variant of SARS-CoV-2 S protein (amino acids 1–1213), a truncated variant that contains the ectodomain of S protein (i.e., recombinant S protein ectodomain) along with a C-terminal Gly-Gly-6xHis tag was subcloned into the pMCAG-2T vector using the GeneArt Type IIs Assembly Kit, *Bbs*I (Thermo Fisher Scientific), according to the manufacturer's instructions. Recombinant SARS-CoV-2 S protein was expressed in Expi293F cells (Thermo Fisher Scientific) as previously described [16]. Five days following transfection, cells were harvested via centrifugation (15,600 x *g*), the supernatant was concentrated and diafiltered using the ÄKTATM flux tangential flow filtration system (Cytiva) into buffer A (10 mM phosphate buffer, 2.7 mM KCl, 500 mM NaCl, pH 8.0). The His-tagged S protein was further purified using Ni-NTA agarose resin (Qiagen), washed with buffer A containing 30 mM imidazole and eluted in buffer A with 200 mM imidazole. Using a Slide-A-Lyzer Dialysis Cassette (20K MWCO, Thermo Fisher Scientific), the eluate was dialysed against PBS (10 mM phosphate buffer, 2.7 mM KCl, 137 mM NaCl, pH 7.5) before use in subsequent experiments.

To measure the cellular immune response from T cells (specifically CD8$^+$ and CD4$^+$ T cells), peripheral blood mononuclear cells (PBMCs) from participants were isolated and tested as described by Shomuradova et al. [16]. In brief, 30 mL of venous blood was collected from each participant and centrifuged via a density gradient (Ficoll; PanEco) for 400 x *g* for 30 min to isolate PBMCs, which were washed with PBS containing 2 mM EDTA. For the enzyme-linked immunosorbent (ELISpot) assay, PBMCs (3 x $10^5$ cells/well) were plated into 96-well nitrocellulose plate that was pre-coated with human IFN-γ capture antibody (ImmunoSpot kit Human IFNγ Single-Color ELISpot kit, Cellular Technology Limited) in serum-free test

medium (Cellular Technology Limited) containing 1 mM GlutaMAX (Gibco) in a final volume of 200 μL/well as previously described [16]. To stimulate the cells, cells were pulsed in duplicates with the S protein at a final concentration of 10 μg/mL or with a pool of overlapping peptides covering the human S protein (PepTivator SARS-CoV-2 Prot S [Cat. No. 130-126-701; Miltenyi Biotec]) at a final concentration of 1 μM. Plates were incubated at 37˚C in 5% $CO_2$ for 18 h, and then the assays were performed according to manufacturer's instructions. Plates were washed twice with PBS, washed twice with PBS + 0.05% Tween-20, and then incubated at room temperature with biotinylated anti-human IFN-γ detection antibody for 2 h. Plates were then washed three times with PBS + 0.05% Tween-20, and then incubated at room temperature with streptavidin-AP for 30 min. After at least two washes, the colorimetric reaction was initiated by adding the substrate components for 15 min at room temperature. The reaction was halted by gently rinsing the plate with distilled water. Spots were counted with the ImmunoSpot Analyzer using the ImmunoSpot software (Cellular Technology Limited). Samples were designated as positive for a T cell response when the mean number of spots in two replicas minus the number of spots in the negative control was ≥10.

Evidence of local and general reactogenicity (frequency and nature of systemic and local immunisation reactions on the day of vaccination and within 7 days after vaccination) was sought on Day 0, Day 2 and Day 7. Adverse events (AEs) were monitored from the day of vaccination (Day 0) onwards, during a scheduled phone call that evening, and at all subsequent clinic visits. A physical examination, including neurologic examination, and vital signs, including body temperature, was conducted at all visits (screening, Day 0, Day 2, Day 7, Day 14, Day 28 and Month 6). Participants received telephone calls after 2, 3, 4 and 5 months, and were asked to answer questions to assess safety and determine if they presented with signs of an acute respiratory infection due to COVID-19. Haematology, clinical biochemistry, coagulation testing and urinalysis were conducted at screening, Day 2 and Day 28; electrocardiography was performed at screening and on Day 2. Immunoglobulin E (IgE) levels were assessed to determine any allergic effects of vaccine components at screening and on Day 28.

## Outcomes

The primary endpoint was the seroconversion rate, specifically the percentage of individuals with a four-fold or higher increase in antibody titres to the RBD of the SARS-CoV-2 S protein, 28 days after vaccination. Secondary endpoints included the assessment of four outcomes: 1) examining the seroconversion rates in response to RBD, S protein and neutralising SARS-CoV-2 antibodies on Day 14, Day 28 (except for RBD) and at Month 6; 2) geometric mean titres (GMTs) of serum antibodies against the RBD, S protein and NAbs on Day 14, Day 28 and Month 6 post-vaccination; 3) GMTs of neutralising antibodies against the Ad5 vector on Day 28 and Month 6 post-vaccination; and 4) the cellular immune response as indicated by the secretion of IFN-γ on Day 14, Day 28 and Month 6 post-vaccination.

Exploratory endpoints were the frequency of laboratory-confirmed COVID-19 cases, severe COVID-19 cases, hospitalisations due to COVID-19 and COVID-19-related deaths (Day 14 to Month 6 post-vaccination). The safety endpoints were reactogenicity (Day 0 to day 7), the frequency and nature of AEs (Day 0 to the end of the study [Month 6 Visit]), and the results of physical and laboratory examinations (i.e., haematological tests, urinalysis, serum IgE concentration).

Of note, two definitions were used to define seroconversion, one quantitative and one qualitative. The primary and secondary endpoints used the quantitative definition: the proportion of participants with at least a four-fold increase in antibody titres against SARS-CoV-2 S protein and/or its RBD, specifically. The qualitative analysis was defined as an antibody titre

above the lower limit of quantification (LLOQ) post-vaccination (if the baseline titre was below the LLOQ), or a four-fold increase over baseline post-vaccination (if the baseline titre was above the LLOQ).

## Statistical methods

Statistical analysis was performed using SPSS Statistics, Version 26.0 (IBM Corp.). An in-depth description of the statistical analysis performed and sample size calculations used in this study is available in S1 Methods. In brief, to guide the data monitoring committee, Pocock boundary interim analysis was performed on data collected for the first 200 volunteers enrolled in the trial up to Day 28 (Visit 5). The applied $p$-value threshold was 0.02616 for the interim analysis set and 0.03039 for the FAS for immunogenicity analysis. Seroconversion rate data was tabulated by evaluation time-points and treatment groups. Intra-group comparisons were performed using the Cochran-Mantel-Haenszel test and a chi-squared test or Fisher's exact test. The 95% confidence intervals (CI) for seroconversion rates were determined using the Clopper-Pearson interval (known as the Exact interval). Antibody titre data were analysed using a two-way analysis of variance (ANOVA), following log-transformation of the data. Titre data are reported as the geometric mean with the 95% lower and upper CIs. Pearson's correlation test was used to determine linear responses between log-transformed baseline anti-Ad5 titres and RBD, S protein or neutralizing SARS-CoV-2 antibodies. Correlations are reported as the mean with the 95% lower and upper CIs. Analysis on the ELISpot assay data was performed using the Mann-Whitney test with Bonferroni adjustment. ELISpot data are presented as the median and quartiles range, with each data point representing the mean of samples performed in duplicate (minus the negative control). The 95% CIs associated with the rate of interferon (IFN)-γ-positivity in participants were determined using the Clopper-Pearson interval. Statistical significance was p<0.05.

## Patient and public involvement

Participants were fully informed about the study and informed consent was obtained. We thank them for their involvement in the study.

## Data sharing

The Authors commit to making the relevant anonymised participant level data available upon reasonable request for 3 years following publication of this study. Requests should be directed to the corresponding author.

# Results

## Participant population

Out of the 783 participants who were screened, this analysis included 500 eligible participants intended for vaccination at six locations in the Russian Federation between 11 September 2020 and 11 November 2020. Of these, 374 participants were randomised to the Ad5-nCoV group and 126 to the Placebo group (Fig 1). The Safety Analysis Set included a total of 496 participants and the FAS for immunogenicity analysis included a total of 495 participants.

The mean age of participants was 41.2 years (range 18–79 years), with 300 (60.5%) participants aged 18–44 years, 161 (32.5%) aged 45–59 years, and 35 (7.1%) aged 60 years or older (Table 1). There were more females (297 out of 496 [59.9%]) than males. The vast majority of the participants (99.4%) were white race and 3 participants were Asian. Baseline characteristics were largely similar across groups.

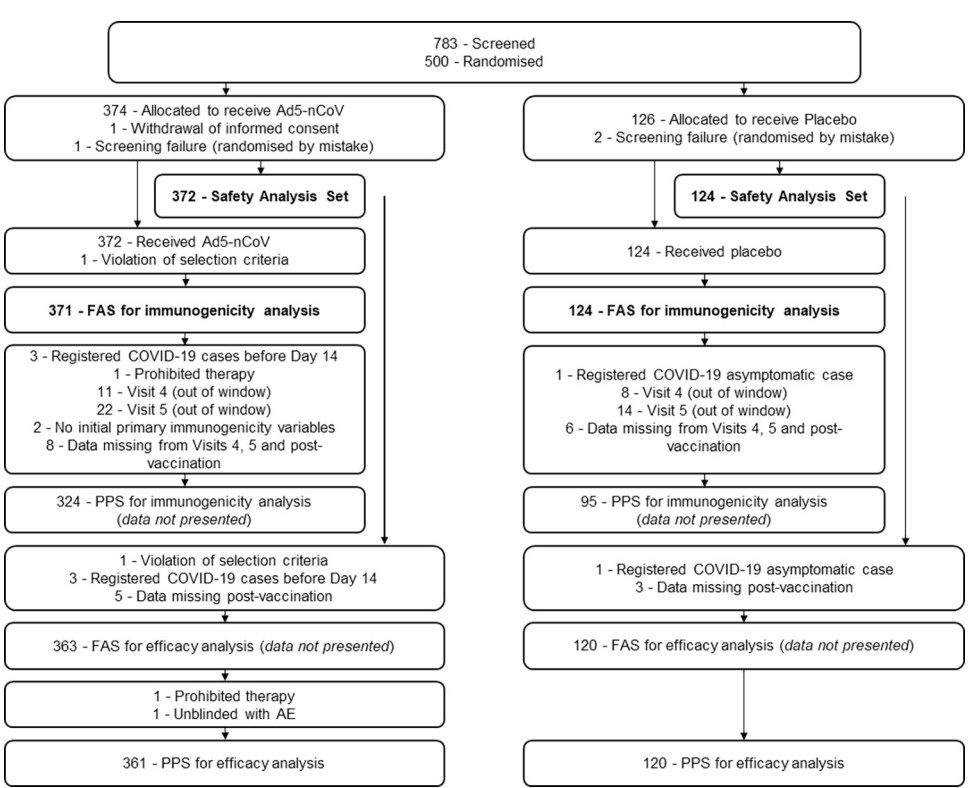

**Fig 1. Participant flow diagram.** Populations used to evaluate the findings of this study are presented in bold text. AE, adverse event; FAS, Full Analysis Set; PPS, Per-Protocol Set.

## Efficacy results

Out of those from the FAS for immunogenicity analysis population, a total of 31 participants were confirmed to have COVID-19 within 6 months post-vaccination (excluding the cases within 14 days post-vaccination). The PPS for efficacy analysis included a total of 481 participants (Fig 1). Within the PPS for efficacy analysis population, 13 cases of COVID-19 occurred in participants from the Placebo group (13/120 [10.83%]), whereas 18 cases of COVID-19 occurred in participants from the Ad5-nCoV group (18/361[4.99%]; p = 0.024). Cases of severe COVID-19 (with the exception of COVID-19 cases that occurred during the first 14 days post-vaccination) occurred in only 2 (1.7%) participants, both of whom were from the Placebo group, and both of these severe COVID-19 cases required hospitalization. Notably, no deaths from COVID-19 occurred during this study. As the data generated were similar between the PPS for efficacy analysis and FAS for immunogenicity analysis populations, only results from the FAS for immunogenicity analysis population were used for the remaining analyses (Fig 1).

## Immunogenicity results

Antibody titres are shown in Fig 2. Baseline antibody titres of the participants in the FAS (Ad5-nCoV; placebo) were similar for anti-RBD, anti-S protein, neutralising anti-SARS-CoV-2 antibodies (NAbs), as well as neutralising antibodies against the Ad5 vector (Fig 2 and S1 Table).

Participants that received the Ad5-nCoV vaccine exhibited a significant immunogenic response against the S protein, with antibody levels significantly increased at Day 14, Day 28 and Month 6 compared with the placebo (p<0.001; Fig 2A and S1 Table). Anti-RBD levels

**Table 1. Participant demographics (safety analysis set).**

|  | Ad5-nCoV N = 372 | Placebo N = 124 | Total N = 496 |
|---|---|---|---|
| **Age (years[%])** |  |  |  |
| 18–44 years | 223 (59.9) | 77 (62.1) | 300 (60.5) |
| 45–59 years | 122 (32.8) | 39 (31.5) | 161(32.5) |
| ≥60 years | 27 (7.3) | 8 (6.5) | 35 (7.1) |
| Mean | 41.2 | 41.0 | 41.2 |
| **Sex (n[%])** |  |  |  |
| Male | 151 (40.6) | 48 (38.7) | 199 (40.1) |
| Female | 221 (59.4) | 76 (61.3) | 297 (59.9) |
| **Race (n[%])** |  |  |  |
| White | 371 (99.7) | 122 (98.4) | 493 (99.4) |
| Asian | 1 (0.3) | 2 (1.6) | 3 (0.6) |
| **Country (n[%])** |  |  |  |
| Russia | 372 (100%) | 124 (100%) | 496 (100%) |
| **Body Mass Index (kg/m$^2$)** |  |  |  |
| Mean | 25.03 | 24.7 | 24.94 |
| Minimum | 18.6 | 18.6 | 18.5 |
| Maximum | 30.0 | 29.9 | 30.0 |
| **Underlying Disease (n[%])** |  |  |  |
| Yes | 98 (26.3) | 29 (23.4) | 127 (25.6) |
| No | 274 (73.7) | 95 (76.6) | 369 (74.4) |
| **Prior Disease (n[%])** |  |  |  |
| Yes | 176 (47.3) | 56 (45.2) | 232 (46.8) |
| No | 196 (52.7) | 68 (54.8) | 264 (54.2) |

were significantly higher at Day 14 and at Day 28 compared with placebo (p<0.001), but after 6 months, GMTs had waned and dropped to levels consistent with those observed in the Placebo group by Month 6 (Fig 2B and S1 Table). Interestingly, a notable increase in anti-S protein and anti-RBD antibodies at Month 6 was also observed in the Placebo group (Fig 2A and 2B and S1 Table), which could be attributed to asymptomatic infections.

Next we examined if the antibodies offered protein against SARS-CoV-2 infection. We found that NAbs were significantly increased by Day 14 in the Ad5-nCoV group compared with levels in the Placebo group, and levels remained significantly higher at 6 Month (p<0.001; Fig 2C and S1 Table). We did observed an approximate two-fold increase in antibody titre levels in the Placebo group by Month 6, which may be indicative of community transmission of SARS-CoV-2. Participants that received the vaccine also exhibited a significant rise in Ad5 antibody levels compared with the Placebo group at Day 28 and at Month 6 (p<0.001; Fig 2D and S1 Table). Notably, NAbs and Ad5 antibodies were also elevated at Month 6 in the Placebo group (Fig 2C and 2D and S1 Table), which could again be attributed to asymptomatic infections or if participants did not report receiving the publically available vaccine that was available in December 2020 [17,18]. Taken together, these data suggest that the Ad5-nCoV vaccine elicits a strong immunogenic response to the S protein, specifically against RBD, with immunity to SARS-CoV-2 infection lasting 6 months.

Seroconversion rates, that is, the primary analysis of the study in which the proportion of participants (out of total participants with seroconversion data) with at least four-fold increase in antibody titres after vaccination, were examined (Fig 3 and S1 Table). For those vaccinated with Ad5-nCoV vs. the placebo, seroconversion rates for S protein, RBD, and NAbs were

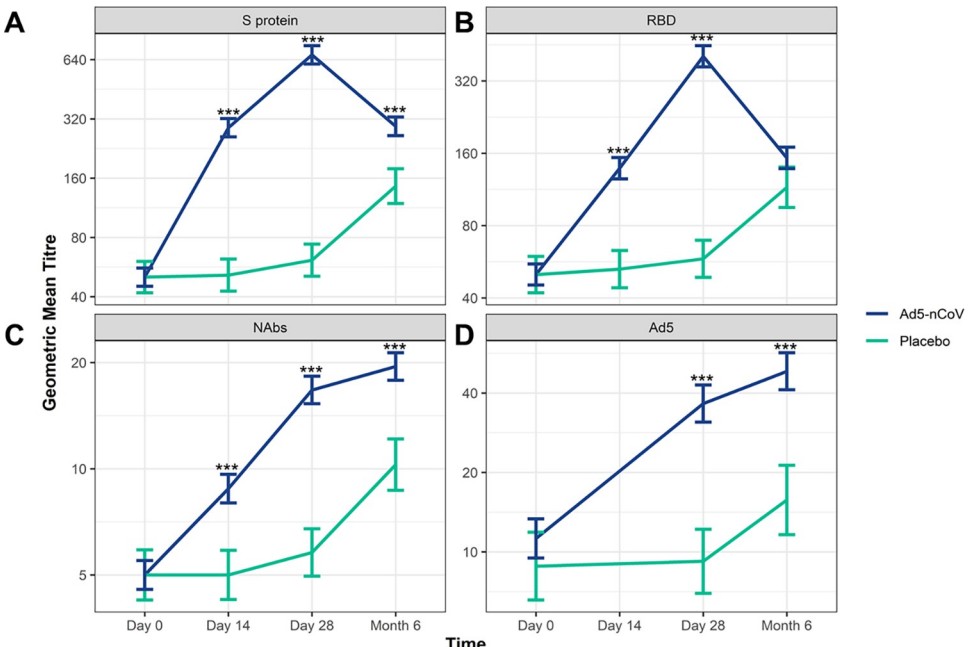

**Fig 2. The Ad5-nCoV vaccine elicits a strong immunogenic response that lasts up to 6 months.** Geometric mean titre (GMT) of serum antibodies against the RBD and SARS-CoV-2 S protein on Day 0, Day 14, Day 28 and Month 6 after vaccination measured from participants from the Ad5-nCoV (n = 371) or the Placebo group (n = 124; from the FAS for immunogenicity analysis). The GMTs with 95% CI are shown for serum antibodies against **A**) S protein, **B**) RBD, **C**) neutralising antibodies against SARS-CoV-2 (NAbs) and **D**) Ad5 neutralising antibodies. Significance between the Ad5-nCoV and Placebo groups was assessed using ANOVA (***, p<0.001). Ad5, adenovirus type-5; ANOVA, analysis of variance; CI, confidence interval; RBD, receptor binding domain; S, spike; SARS-CoV-2, severe acute respiratory syndrome coronavirus 2.

significantly elevated by Day 14 (p<0.001; Fig 3A–3C and S1 Table), and the seroconversion rates peaked at Day 28 for S protein (90.6% [95% CI: 87.2, 93.4] vs. 6.72% [95% CI: 2.95, 12.8]), RBD (78.5% [95% CI: 73.9, 82.6] vs. 5.88% [95% CI: 2.4, 11.7]), and NAbs (59.0% [95% CI: 53.3, 64.6] vs. 3.88% [95% CI: 1.07, 9.65]) (p<0.001). Seroconversion rates for S protein and NAbs in the Ad5-nCoV group were elevated after 6 months compared with the Placebo group (p<0.001 and p<0.05, respectively; Fig 3A and 3C, and S1 Table), indicating that some protection remained at the later timepoint. For those vaccinated with Ad5-nCoV, the seroconversion rates against anti-Ad5 neutralising antibodies were significantly higher at Day 28 and Month 6 compared with the Placebo (p<0.001; Fig 3D and S1 Table).

Next, pre-existing anti-Ad5 neutralising antibody titres were examined between participants in the Ad5-nCoV and Placebo groups at Day 0, Day 28 and Month 6 post-vaccination. Almost all participants had neutralising antibodies against the Ad5 vector at baseline: 317/371 (85.5%) participants who received Ad5-nCoV and 106/124 (85.5%) who received placebo. Of the total participants with pre-existing anti-Ad5 titres at Day 0 (n = 423), most participants had low Ad5 antibody titres (≤1:200), whereas only seven participants (1.7%) with pre-existing anti-Ad5 antibodies had high levels (>1:200). Of those in the Ad5-nCoV group, GMTs of anti-Ad5 participants with low pre-existing anti-Ad5 antibodies increased from 11.3 at baseline (n = 317) to 36.5 at Day 28 (n = 353), which then increased further to 48.3 (n = 355) at Month 6. Corresponding values for the very few participants with pre-existing immunity to Ad5 >1:200 showed an increase from 320.0 at baseline (n = 6) to 359.2 at Day 28 (n = 6) and a further increase to 557.2 at Month 6 (n = 6).

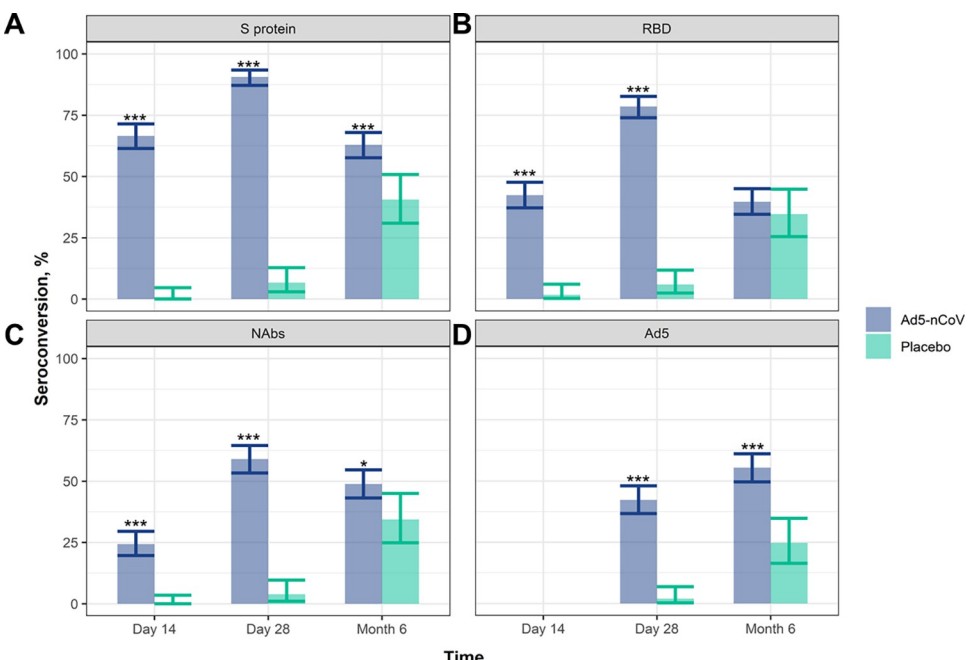

**Fig 3. High seroconversion rates persist up to 6 months post-vaccination.** Seroconversion rates were examined on Day 14, Day 28 and Month 6 post-vaccination measured from participants in the Ad5-nCoV (n = 371) or the Placebo group (n = 124; from the FAS for immunogenicity analysis). The seroconversion rates with 95% CI (calculated using the Clopper-Pearson method) are shown for serum antibodies against **A**) S protein, **B**) RBD, **C**) neutralising antibodies against SARS-CoV-2 (NAbs) and **D**) Ad5 neutralising antibodies. Significance between the Ad5-nCoV and Placebo groups was assessed using ANOVA (***, p<0.001; *, p<0.05). Ad5, adenovirus type-5; ANOVA, analysis of variance; CI, confidence interval; RBD, receptor binding domain; S, spike; SARS-CoV-2, severe acute respiratory syndrome coronavirus 2.

As illustrated in Fig 4, to determine the distribution of anti-Ad5 antibodies, participants in each group were separated based on their pre-existing anti-Ad5 neutralising antibodies as low (≤1:200; i.e., 1:5 through 1:160) or high (>1:200; i.e., 1:320 through 1:1280). The majority of participants at Day 0 had low anti-Ad5 titres with similar distributions between the Ad5-nCoV and Placebo groups. At Day 28 post-vaccination, those in the Ad5-nCoV group had higher titre levels of anti-Ad5 as indicated by a shift in titre distribution to the right of the threshold (dashed vertical line) intercepting the low and high titre levels (Fig 4). Higher anti-Ad5 distribution levels persisted at Month 6 post-vaccination (Fig 4). At Day 28 post-vaccination, those in the Placebo group had a similar distribution of anti-Ad5 antibodies as observed at Day 0; however, a slight increase in a shift towards higher anti-Ad5 titres was noted at Month 6 (Fig 4), which may be attributed to participants that did not report receiving the publically available vaccine (at the time this included Sputnik V, which used recombinant Ad26 and Ad5 vectors for the first and second doses, respectively) [17,18]. These data demonstrate an increase in anti-Ad5 neutralising antibodies that persists for up to 6 months post-vaccination.

The effect of pre-existing immunity to anti-Ad5 neutralising antibodies on the immunological responses to the vaccine was studied. Participants with pre-existing anti-Ad5 antibodies that had received the Ad5-nCoV vaccine (n = 317) were separated into seven subgroups: six subgroups had low (≤1:200) Ad5 titres and one subgroup had high (>1:200) Ad5 titres and one subgroup had high (>1:200) Ad5 titre (Fig 5). Participants with the lowest amount of pre-existing anti-Ad5 titres (subgroup 1:5) exhibited the highest peak in GMTs against S protein (Fig 5A) and RBD (Fig 5B) at Day 28 post-vaccination. All other subgroups, including those

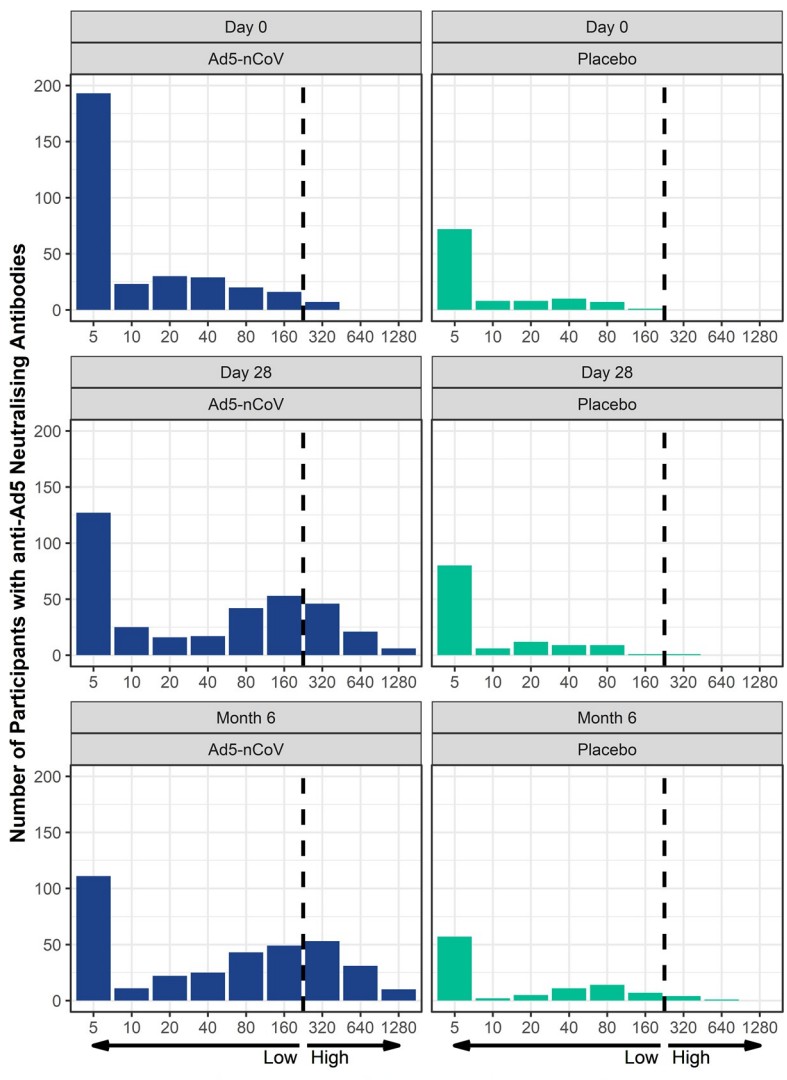

**Fig 4. Titres to anti-Ad5 neutralising antibodies increase and persist up to 6 months post-vaccination.** Participants were from the FAS for immunogenicity analysis population (Ad5-nCoV, n = 371; Placebo group, n = 124). Based on the GMTs of anti-Ad5 on Day 0, Day 28 and Month 6, the number of participants with low (≤1:200) or high (>1:200) anti-Ad5 titres those from either the Ad5-nCoV (left panels) or Placebo groups (right panels) is presented. Participants in each group were separated based upon their pre-existing anti-Ad5 neutralising antibodies as low (≤1:200; i.e., 1:5, 1:10, 1:20, 1:40, 1:80, 1:160) or high (>1:200; i.e., 1:320, 1:640, 1:1280). The threshold for low and high anti-Ad5 titres is indicated with the dashed vertical line.

with the highest anti-Ad5 levels (subgroup 1:320; n = 7), showed comparable GMTs against S protein and RBD at Day 28 (Fig 5A and 5B). However, by 6 months post-vaccination, GMTs against S protein and RBD continued to increase in those with the highest anti-Ad5 titres (subgroup 1:320), whereas the GMTs in all other subgroups began to decline at 6 months (Fig 5A and 5B). Geometric mean titres for NAbs (Fig 5C) and Ad5 (Fig 5D) varied amongst all subgroups with pre-existing low antibody titres at Day 28 and by Month 6. However, only participants with pre-existing high Ad5 antibody titres (subgroup 1:320) continued to show high GMTs for NAbs and Ad5 at 6 months (Fig 5C and 5D). Additionally, GMTs for Ad5 were proportional to the level of pre-existing anti-Ad5 neutralising antibodies, i.e., higher pre-existing

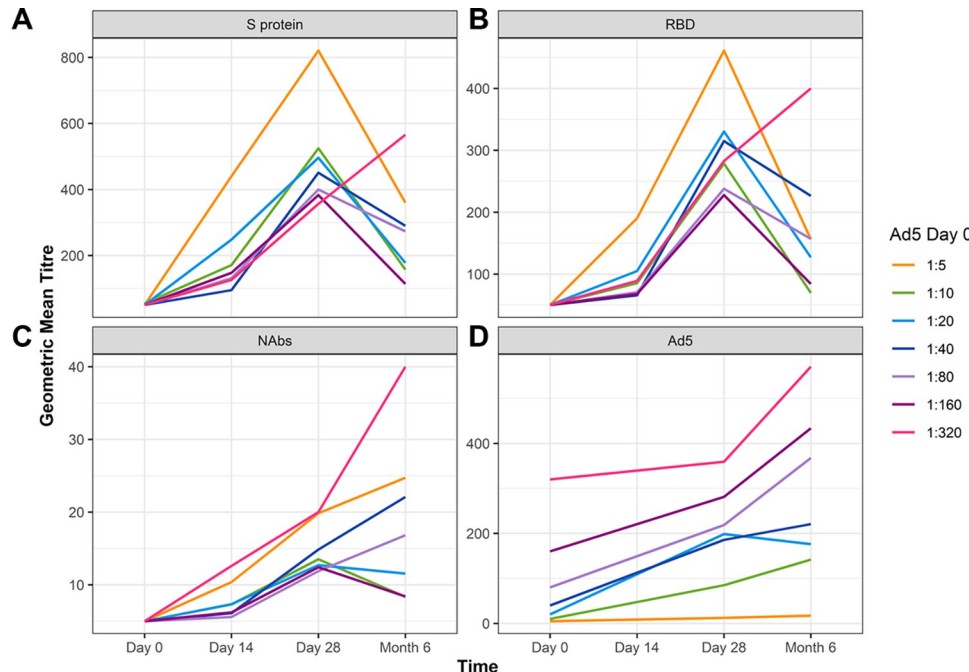

**Fig 5. Antibody responses to SARS-CoV-2 vaccine peak at Day 28 in those with low pre-existing immunity to Ad5.** Geometric mean titres in response to serum antibodies **A**) S protein, **B**) RBD, **C**) neutralising antibodies against SARS-CoV-2 (NAbs) and **D**) Ad5 neutralising antibodies were examined on Day 0, Day 14, Day 28 and Month 6 after vaccination. Participants from the FAS for immunogenicity analysis population that received the Ad5-nCoV vaccine (n = 371) were divided into seven subgroups (from 1:5 to 1:320) based on their pre-existing anti-Ad5 antibody levels. Ad5, adenovirus type-5; RBD, receptor binding domain; S, spike; SARS-CoV 2, severe acute respiratory syndrome coronavirus 2.

anti-Ad5 antibodies lead to higher titre levels post-vaccination (Fig 5D). Collectively, these data suggest that an established pre-existing immunity to Ad5, as determined by the presence of low antibody titre levels, elicits a short-term immune response post-vaccination to the S protein and RBD antigens that declines after 28 days. However, participants with higher pre-existing Ad5 antibody titres had a sustained immunogenic response to these antigens, producing NAbs up to 6 months post-vaccination.

Pearson's correlation coefficient was calculated to correlate serological responses to Ad5 load, that is, baseline GMTs to Ad5 (i.e., pre-existing Ad5 antibodies) were correlated against GMTs to RBD, S protein and NAbs post-vaccination (Fig 6). There was a negative correlation of -0.36 between baseline GMT to Ad5 and GMT to RBD on Day 14 that was +0.05 at 6 months post-vaccination. The correlation magnitudes of GMTs to S protein and neutralising SARS-CoV-2 antibodies decreased from Day 14 to Month 6 (Fig 6). These changes in GMT correlations indicate that the relationship between levels of baseline Ad5 antibodies to the COVID-19 humoral immune response diminishes over time following vaccination.

The effect of the vaccination on cellular immune responses was studied in a subgroup of 69 participants recruited by the Moscow clinic site (Ad5-nCoV group, n = 50 volunteers; Placebo group, n = 19). The presence of SARS-CoV-2 specific CD8[+] and CD4[+] T cells in venous blood samples was assessed by counting the number of spot-forming cells (SFCs; i.e., IFN-γ secreting cells) in an ELISpot assay in which isolated PBMCs were stimulated with either recombinant S protein ectodomain or a pool of overlapping peptides covering the S protein (i.e., peptide pool). The most pronounced response occurred on Day 14 after the Ad5-nCoV vaccination in which the median number of SFCs was 32.8 (Quartile [Q]1: 19.0, Q3: 78.9) when stimulated

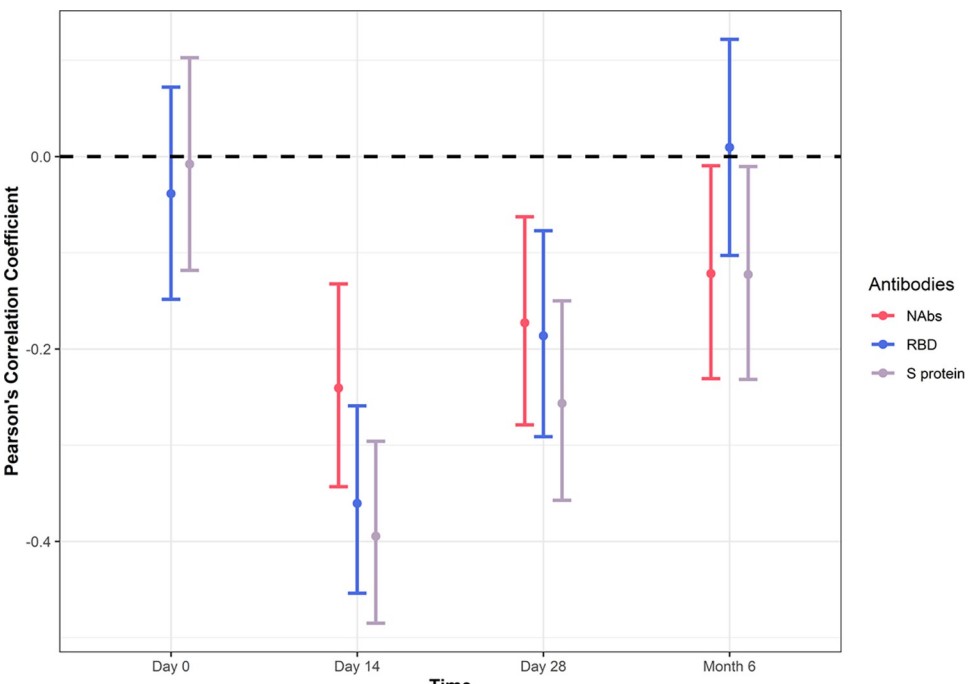

**Fig 6. Immunity to RBD of SARS-CoV-2 diminishes by 6 months post-vaccination in those with pre-existing Ad5 antibodies.** Pearson's correlation coefficient between GMTs versus anti-Ad5 antibodies (ADENOVAB) in response to anti-RBD and anti-S protein antibodies as well as neutralising SARS-CoV-2 antibodies (NAbs) on Day 0, Day 14, Day 28 and Month 6 after vaccination. Participants were from the FAS for immunogenicity analysis population that received the Ad5-nCoV vaccine and had pre-existing Ad5 antibodies (n = 317). The correlation coefficients with 95% CI are shown. Ad5, adenovirus type-5; CI, confidence interval; RBD, receptor binding domain; S, spike; SARS-CoV 2, severe acute respiratory syndrome coronavirus 2.

with the peptide pool and 32.2 (Q1: 16.0, Q3: 99.9) when stimulated with recombinant S protein ectodomain (Fig 7A and 7B, respectively). For cells stimulated with the peptide pool, the median number of SFCs decreased to 5.5 (Q1: 1.5, Q3: 12.2) on Day 28 and further still to 1.5 (Q1: 0, Q3: 8.5) at Month 6. When stimulated with recombinant S protein ectodomain, the number of SFCs increased to 4.5 (Q1: 3.0, Q3: 12.5) on Day 28 and 5.5 (Q1: 0.75, Q3: 13.8) by 6 months post-vaccination.

Compared with the response observed in the Placebo group, those in the Ad5-nCoV group displayed significantly higher numbers of IFN-γ-positive T cells, regardless of how the cells were stimulated, on Days 14 (p<0.001) and 28 (p<0.01). However, 6 months post-vaccination, the differences between the groups were not statistically significant when cells were stimulated with the recombinant S protein ectodomain (p = 0.6) or with the peptide pool (p = 1.0).

As shown in Table 2, the percentage of participants who had an IFN-γ-positive response above the threshold following stimulation with the peptide pool on Day 14 in the Ad5-nCoV group was 91.7% (95% CI: 80.0; 97.7), whereas no participants were positive in the Placebo group. During the follow-up period, the response rate decreased for those in the Ad5-nCoV group and the percentage of patients that displayed an IFN-γ response on Day 28 was 37.2% (95% CI: 23.0; 53.3), whereas no participants were positive in the Placebo group. At Month 6, the percentage of patients displayed an IFN-γ response was 21.3% and 12.5% in the Ad5-nCoV and Placebo groups, respectively.

There was a similar trend in IFN-γ-positive responses when PBMCs were stimulated with recombinant S protein ectodomain (Table 2). The percentage of participants who displayed an

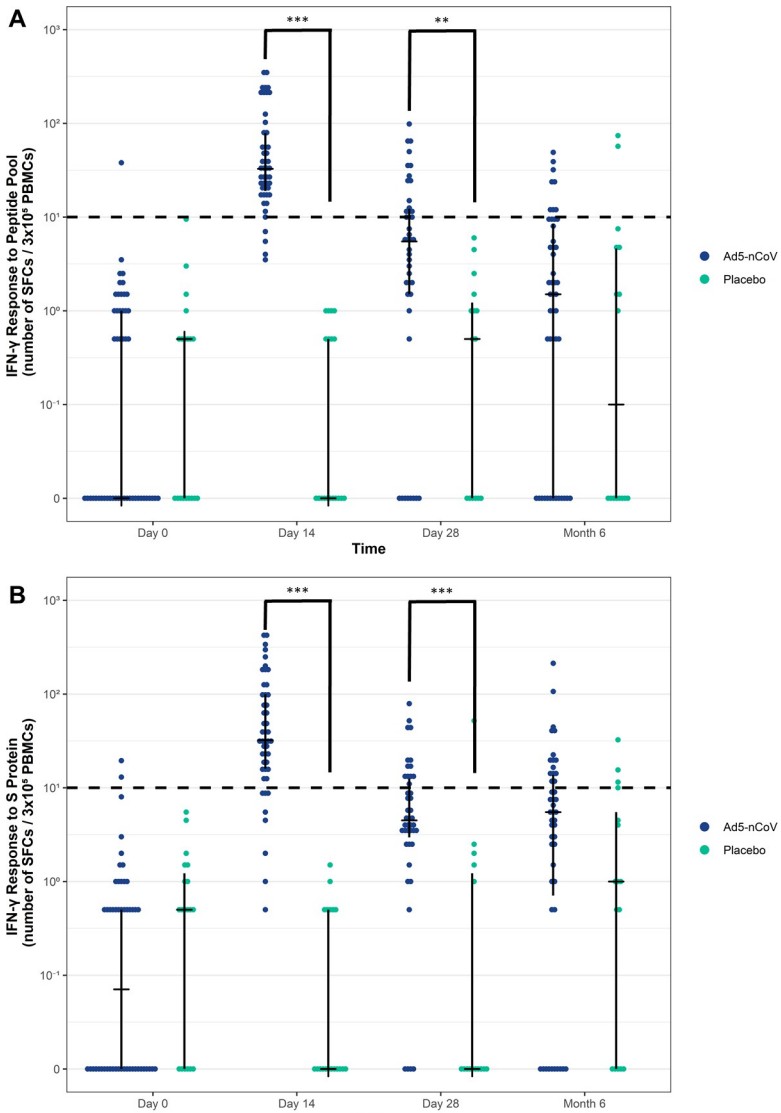

**Fig 7. The Ad5-nCoV vaccine induces a durable cellular immune response.** The cellular immune response (IFN-γ-positive cells from isolated PBMCs) was evaluated following the SARS-CoV-2 vaccine using an enzyme-linked immunospot (ELISpot) assay. Peripheral blood mononuclear cells were isolated from blood samples taken from each participant in the Placebo (n = 19) and Ad5-nCoV (n = 50) groups. Participants were a subset from the FAS for immunogenicity analysis population. Isolated PBMCs were seeded at 3 x10$^5$ cells/well in a 96-well plate on Day 0, Day 14, Day 28 and Month 6 after vaccination. Cells were stimulated with (**A**) a pool of peptides that span the S protein (i.e., peptide pool) or (**B**) the recombinant S protein ectodomain (i.e., S protein). Interferon-γ-positive spot-forming cells (SFCs) were quantified via an automated digital image. Samples from each participant were analyzed in duplicate and the mean of two values (minus the negative control) represents each dot. The median with quartiles is the plotted, the dashed line indicates the positivity threshold of 10$^1$ SFCs. For group comparisons, the Mann-Whitney test with Bonferroni adjustment was used (***, p<0.001; **, p<0.01). Ad5, adenovirus type-5; IFN-γ, interferon-γ; PBMC, peripheral blood mononuclear cell; S, spike; SARS-CoV-2, severe acute respiratory syndrome coronavirus 2.

IFN-γ response in the Ad5-nCoV group on Day 14 was 83.3% (95% CI: 69.8; 92.5), whereas no participants were positive in the Placebo group. During the follow-up period, the response rate markedly decreased for those in the Ad5-nCoV group and the percentage of participants that displayed an IFN-γ response on Day 28 was 30.2% (95% CI: 17.2; 46.1). At Month 6, 31.9% of participants displayed an IFN-γ response that was comparable to 25.0% of participants in the

**Table 2. Summary of participants with IFN-γ-positive samples following vaccination.**

| Stimulant | Parameter | Ad5-nCoV N = 50 | | | Placebo N = 19 | | | |
|---|---|---|---|---|---|---|---|---|
| | | Day 14 | Day 28 | Day 14 | Day 28 | Day 14 | Day 28 | |
| S protein Peptide Pool | Samples (n/N) | 44/48 | 16/43 | 10/47 | 0/19 | 0/15 | 2/16 | |
| | IFN-γ⁺, % (95% CI) | 91.7 (80.0, 97.7) | 37.2 (23.0, 53.3) | 21.3 (10.7, 35.7) | 0 | 0 | 12.5 (1.55, 38.3) | |
| Recombinant S Protein ectodomain | Samples (n/N) | 40/48 | 13/43 | 15/47 | 0/19 | 1/15 | 4/16 | |
| | IFN-γ⁺, % (95% CI) | 83.3 (69.8, 92.5) | 30.2 (17.2, 46.1) | 31.9 (19.1, 47.1) | 0 | 6.67 (0.169, 31.9) | 25.0 (7.27, 52.4) | |

Ab, antibody; Ad5-nCoV, adenovirus type-5 vectored COVID-19 vaccine; CI, confidence interval; COVID-19, coronavirus disease 2019; FAS, full analysis set; N, total number of participants; n, number of participants with IFN-γ-positive samples; IFN-γ⁺, interferon- γ -positivity.

Notes: Participants were a subset of the FAS for immunogenicity analysis population that visited the Moscow clinic site.

The rate of IFN-γ⁺ was based on the positivity of samples performed in duplicate, with samples collected from each participant.

The 95% CIs were determined using the Clopper-Pearson interval (also known as the Exact interval).

Placebo group following stimulation with recombinant S protein ectodomain. As previously referenced, the increase in the Placebo group at Month 6 could be attributed to asymptomatic infections or participants that did not report receiving the publically available vaccination. Nonetheless, these combined data indicate that the Ad5-nCoV vaccine generates durable SARS-CoV-2–specific immunity.

## Safety evaluation

**Systemic (general) immunisation reactions.** A total of 100 (26.9%) of the 372 participants who received the Ad5-nCoV vaccine reported systemic reactions (Table 3). The incidence of systemic reactions was significantly higher than in the Placebo group, where 13 (10.5%) of the 124 participants reported systemic reactions.

The most commonly reported reactions in the Ad5-nCoV group included increase in body temperature (20.2%), headache (5.9%), fatigue (5.4%), myalgia (4.8%) and arthralgia (1.9%; S2 Table). For most participants, general immunisation reactions were mild (Grade 1, 21.0%). Moderate (Grade 2) reactions occurred in 4.6% of participants and included increase in body temperature, headache, myalgia, arthralgia, and fatigue. There were a total of five severe (Grade 3) reactions, including an increase in body temperature (39.0–40.0˚C), occurred in 4 participants, and myalgia occurred in only 1 participant (Tables 3 and S3).

In the Placebo group, reactions included increase in body temperature (6.5%), headache (4.8%), fatigue (4.8%) and diarrhoea (0.8%). These AEs were mild or moderate; no severe reactions were reported (Tables 3, S2 and S3).

There were no fatal outcomes. During the first 7 days after vaccination, a total of six serious AEs (SAEs) were reported (Table 3): one SAE in 1 (0.3%) participant from the Ad5-nCoV group and five SAEs in 5 (4.0%) participants from the Placebo group. Of note, two of the participants from the Placebo group were hospitalised due to COVID-19. For the only SAE event experienced by a participant from the Ad5-nCoV group, the event was found to have no connection with the study drug. The SAEs are described in S4 Table.

**Injection site reactions.** Injection site reactions occurred in 106 participants (28.5%) who received the Ad5-nCoV vaccine. By comparison, only 2 participants in the Placebo group (1.6%) experienced an injection site reaction including swelling and pain or induration (S5 Table), all of which were Grade 1 in severity (S6 Table). The most common reactions in the vaccine group included pain (16.9%), erythema (14.8%) and induration at the injection site

**Table 3. Summary of adverse events (AEs) by safety analysis set.**

| | Ad5-nCoV N = 372 n (%) | Placebo N = 124 n (%) | Total N = 496 n (%) |
|---|---|---|---|
| **Systemic (general) immunisation reactions** | **100/162 (26.9)** | **13/20 (10.5)** | **113/182 (22.8)** |
| Grade 1 | 78/131 (21.0) | 10/16 (8.1) | 88/147 (17.7) |
| Grade 2 | 17/26 (4.6) | 3/4 (2.4) | 20/30 (4.0) |
| Grade 3 | 5 (1.3) | 0 | 5 (1.0) |
| **Local immunisation reactions** | **106/180 (28.5)** | **2/4 (1.6)** | **108/184 (21.8)** |
| Grade 1 | 91/154 (24.5) | 2/4 (1.6) | 93/158 (18.8) |
| Grade 2 | 15/26 (4.0) | 0 | 15/26 (3.0) |
| Grade 3 | 0 | 0 | 0 |
| **AEs (except for immunisation reactions)** | **152/365 (40.9)** | **38/75 (30.6)** | **190/440 (38.3)** |
| **AEs related to vaccination** | **130/280 (34.9)** | **25/43 (20.2)** | **155/323 (31.3)** |
| Grade 1 | 106/246 (28.5) | 20/36 (16.1) | 126/282 (25.4) |
| Grade 2 | 21/30 (5.6) | 5/7 (4.0) | 26/37 (5.2) |
| Grade 3 | 3/4 (0.8) | 0 | 3/4 (0.6) |
| **AEs related to vaccination and registered during the first 7 days after vaccination** | **123/244 (33.1)** | **25/42 (20.2)** | **148/286 (29.8)** |
| Serious AEs | 1/1 (0.3) | 5/5 (4.0) | 6/6 (1.2) |
| Death | 0 | 0 | 0 |
| Early discontinuation due to AE | 0 | 0 | 0 |

AE, adverse event.

For AEs at the indicated severity with two numbers, n represents the number of participants per category/number of AEs as one participant experienced several cases of the same AE during the study.

(3.8%; S5 Table). As listed in S6 Table, for most participants, local injection site reactions were mild (24.5%). While moderate reactions rarely occurred in those that received the Ad5-nCoV vaccine, 15 participants (4%) in the Ad5-nCoV group experienced the following local post-vaccination reactions of moderate severity at the injection site: erythema (8 [2.2%]), pain (6 [1.6%]), swelling (5 [1.3%]), and induration (1 [0.3%]). No severe reactions were reported.

**Other adverse events (excluding immunisation reactions).** As listed in Table 3, a total of 190 participants (38.3%) reported 440 AEs during the 6 months after vaccination, of which 152 (40.9%) participants from the Ad5-nCoV group reported 365 AEs and 38/75 (3.6%) participants from the Placebo group reported 75 AEs. These AEs were assessed by the investigators as related to the vaccine (Ad5-nCoV: 34.9%; Placebo: 20.2%; Table 3). The majority of vaccine-related AEs were reported during the first 7 days after vaccination (Ad5-nCoV: 33.1%; Placebo: 20.2%). In both groups, the most common AEs were out-of-range laboratory measurements: 34.4% reported by participants in the Ad5-nCoV group and 16.9% from the Placebo group. The most pronounced differences in AE incidence in the Ad5-nCoV group compared with the Placebo group were: increases in C-reactive protein (18.0% vs. 2.4%), monocytes (4.0% vs. 0%) and aspartate aminotransferase (2.4% vs. 1.6%) as well as a decrease in neutrophils (6.5% vs 2.4%) (S7 Table).

For most participants, AEs were mild, reported in 28.5% of the Ad5-nCoV group and 16.1% of the Placebo group. Moderate events were registered in 5.6% of participants who received the Ad5-nCoV vaccine and 4.0% who received the placebo. By comparison, SAEs occurred in 0.8% of participants in the Ad5-nCoV group, but none in the Placebo group; these included increases in C-reactive protein (0.5%) and blood fibrinogen levels (0.5%).

For most participants, general and injection site reactions and AEs resolved within 7 days after vaccination.

Results of the analysis of laboratory parameters demonstrated a trend towards an increase in C-reactive protein levels, an increase in the red cell sedimentation rate, an increase in the mean percentage of monocytes, and a decrease in the mean percentage of neutrophils after administration of Ad5-nCoV (S7 Table). The changes in the examined laboratory parameters were pronounced the day after vaccination, but had largely resolved by Day 28.

The proportion of participants in the Ad5-nCoV group with elevated IgE at Day 28 showed no changes from screening: 73/372 (19.6%) to 76/365 (20.8%).

## Discussion

In this multicentre, randomised, double-blind, placebo-controlled, phase 3 trial including 500 adult participants aged 18–85 years (mean age: 41.2 years), the immunogenicity, efficacy and safety of the Ad5-nCoV COVID-19 vaccine was assessed up to 6 months after vaccination. Both study groups (Ad5-nCoV and Placebo) had similar baseline characteristics. A single immunisation with the Ad5-nCoV vaccine led to a marked immune response. The primary endpoint (seroconversion rate of anti-RBD antibodies on Day 28 after vaccination) and all secondary endpoints on Day 28 after vaccination showed a strong immunogenic response to the Ad5-nCoV vaccine.

The Ad5-nCoV vaccine has a good safety profile comparable with the findings of preceding clinical trials in healthy adults. Most post-vaccination AEs were mild or moderate in severity. Although the proportions of participants who reported adverse reactions such as an increase in body temperature, headache, and pain at the injection site were higher in those that received the Ad5-nCoV vaccine, adverse reactions within 28 days were generally mild to moderate and the majority resolved within 7 days after vaccination. All Grade 3 AEs occurred among participants from the Ad5-nCoV group and were similar to commonly reported AEs after other types of immunisation.

For those vaccinated with Ad5-nCoV, the seroconversion rate of antibodies against RBD (78.5%) and of NAbs (59.0%) on Day 28 (Fig 3 and S1 Table) were similar to the results obtained during the interim analysis of a phase 2 clinical study conducted in China. The seroconversion rate of anti-RBD antibodies in the Ad5-nCoV group was 97% (95% CI: 92; 99), and the seroconversion rate of neutralising antibodies against live SARS-CoV-2 was 47% (95% CI: 39; 56) [9].

Interestingly, GMTs of anti-RBD, S protein-specific antibodies and neutralizing SARS-CoV-2 antibodies were elevated at Month 6 post-vaccination in the Placebo group, not by Day 14 or Day 28 (Fig 2 and S1 Table). We theorized two possible explanations. First, some individuals might have been asymptomatic while infected with COVID-19, or COVID-19-positive patients did not report their symptoms to the study's physicians. Second, individuals in the Placebo group might have taken a commercially available antibody test that was widely available in Russia and decided to receive another vaccination after observing their low antibody titres, thereby confounding their treatment status. During the course of the trial, the COVID-19 vaccine, Sputnik V, became widely available in December 2020, and the vaccination regimen uses recombinant Ad26 and Ad5 vectors for the first and second doses, respectively [17,18]. Unreported vaccinations amongst those in the Placebo group would also explain the increase in anti-Ad5 titres observed at Month 6 post-vaccination (Fig 2).

Even disregarding other COVID-19 candidate vaccines that use a recombinant Ad5 vector, adenovirus exposure itself is common. Pre-existing anti-Ad5 immunity may affect the immunogenicity of Ad5-based vaccines, and as a result, their efficacy against COVID-19. However, the proportion of individuals with high anti-Ad5 titres can vary across geographical regions [19–21] and it is unclear to what extent previous exposure influences existing titres and the

speed of their decline. A titre of 1:200 was selected as the cut-off point for low and high anti-Ad5 antibodies during the phase 1 and 2 studies in China [9,10]. Those with low baseline anti-Ad5 titres (≤1:200) had RBD-specific antibody and neutralising antibody levels roughly twice as high as those with high baseline anti-Ad5 titres (>1:200) [9,10]. Our data analysis used a cut-off of 1:200 in line with the previous Chinese studies [9,10], but given the variable levels of anti-Ad5 titres around the world, a limit of 1:200 may not necessarily be appropriate in all regions. The selection of an appropriate cut-off point in the current study was hampered by a lack of published data on anti-Ad5 levels in the Russian population. Among the Russian participants, seven (1.4%) participants presented with high baseline levels of anti-Ad5 antibodies (1:320; Fig 4). All of them were in the subgroup of participants aged 18 to 45 years.

In this study, we also uncovered the response of pre-existing immunity to Ad5 and showed how it affects subsequent humoral immune responses as well as the longevity of immunity to COVID-19. In participants vaccinated with Ad5-nCoV, there was no difference in humoral immunity to SARS-CoV-2 between those with pre-existing anti-Ad5 titres at baseline above or below the cut-off of 1:200. However using a cut-off of 1:5, we showed that those with pre-existing low Ad5 titres (1:5; Fig 5) displayed a greater amount of GMTs to anti-RBD and S protein antibodies as well as NAbs by Day 28 than those with a higher baseline ratio of Ad5 titres (>1:200). The relationship between the levels of baseline Ad5 GMTs to the respective antigen tends to weaken over time following vaccination (Fig 6). Studies with longer follow-up periods would help to clarify the longevity of the immune response in those vaccinated with Ad5-nCoV, particularly with multiple COVID-19 variants circulating amongst the global population. Moreover, studies with a larger dataset would help to determine if pre-existing immunity to Ad5 correlates to protection efficacy.

Our study provides data on the incidence of anti-Ad5 immunity in the Russian population, admittedly in a relatively small, sample size. Almost all participants had pre-existing neutralising antibodies to Ad5, although levels were generally very low. No such data were available before this study, and only limited and highly variable data are available for the level of Ad5 immunity in Europe.

There are currently multiple recombinant Ad-vectored COVID-19 candidate vaccines registered or in development [4–7,9,10,22,23]. The most relevant comparator vaccine in terms of the practicalities of storage, transport and administration is the Ad26.COV2.S vaccine [24]. Like the Ad5-nCoV vaccine, it can be administered as a single dose, stored in a standard refrigerator without the need for ultra-low temperature freezing and is stable at room temperature prior to administration. An interim analysis of Ad26.COV2.S showed that vaccine-induced neutralising antibodies against wild-type virus were detected in 95% or more of participants on Day 29 after dosing. Our qualitative analysis used a similar definition of seroconversion to that used to assess the Ad26.COV2.S vaccine [24]; as well as including participants who seroconverted from below the LLOQ of the assay at baseline to detectable antibodies at Day 28, it included those with a four-fold or greater increase in antibody titre over the same period. In a previous study, the Ad5-nCoV vaccine provided a similar antibody response to Ad26.COV2.S in which 95.9% of participants developed antibodies to the S protein and 92.5% to the smaller anti-RBD region by Day 28 [24].

Other study limitations included the fact all participants were white, although conversely, this also provided the first data in a non-Chinese, European population. Nor did this trial include children or pregnant women, and there were only a small number of older adults (35 were ≥60 years in the Ad5-nCoV group). An ideal candidate vaccine for COVID-19 should cover vulnerable populations of all ages. Anti-S protein-specific antibodies have been reported to decline rapidly in individuals who have recovered from COVID-19, especially those who were asymptomatic or had mild symptoms [25]. Additionally, the sample size was relatively

small and some of the calculated 95% CIs were wide. Finally, virus mutation, an emerging problem, may reduce the effectiveness of current vaccines [26]. It is not known whether participants of this study were exposed to any COVID-19 variants. Further study is underway to determine neutralising antibodies to the widely circulating variants following vaccination with Ad5-nCoV, which include but are not limited to the Alpha (B.1.1.7), Beta (B.1.351) and Gamma (P.1) variants.

## Conclusions

Analysis of data from this phase 3 trial demonstrated the immunogenicity and safety of this Ad5-vector based COVID-19 vaccine. More data are required to determine whether this vaccine reduces infections and transmission. Overall, this stable, single-dose vaccine could contribute to the global fight against the evolving SARS-CoV-2 virus.

## Supporting information

**S1 Checklist. CONSORT checklist.**
(DOC)

**S1 Protocol. Study protocol.**
(DOCX)

**S1 Methods. Additional details of statistical analysis.**
(DOCX)

**S1 Table. Summary of geometric mean titres and seroconversion rates post-vaccination with Ad5-nCoV or placebo (full analysis set).**
(DOCX)

**S2 Table. Systemic (general) post-vaccination reactions (safety analysis set).**
(DOCX)

**S3 Table. Systemic (general) post-vaccination reactions by severity (safety analysis set).**
(DOCX)

**S4 Table. Summary of all severe adverse events that led to hospitalisation or prolonged hospitalisation (safety analysis set).**
(DOCX)

**S5 Table. Local post-vaccination reactions (safety analysis set).**
(DOCX)

**S6 Table. Local post-vaccination reactions by severity (safety analysis set).**
(DOCX)

**S7 Table. Other adverse events unrelated to vaccination reactions (safety analysis set).**
(DOCX)

## Author Contributions

**Conceptualization:** Dmitry Lioznov, Irina Amosova, Savely A. Sheetikov, Ksenia V. Zornikova, Yana Serdyuk, Grigory A. Efimov, Mikhail Khmelevskii, Andrei Afanasiev, Nadezhda Khomyakova, Dmitry Zubkov, Anton Tikhonov, Tao Zhu, Vitalina Dzutseva.

**Data curation:** Irina Amosova, Andrei Afanasiev.

**Formal analysis:** Ksenia V. Zornikova, Grigory A. Efimov, Luis Barreto.

**Funding acquisition:** Mikhail Tsyferov.

**Investigation:** Ksenia V. Zornikova, Grigory A. Efimov, Andrei Afanasiev, Anton Tikhonov.

**Methodology:** Irina Amosova, Savely A. Sheetikov, Ksenia V. Zornikova, Yana Serdyuk, Dmitry Zubkov, Anton Tikhonov, Luis Barreto.

**Project administration:** Yana Serdyuk.

**Resources:** Mikhail Tsyferov.

**Software:** Mikhail Tsyferov, Andrei Afanasiev.

**Supervision:** Mikhail Tsyferov, Nadezhda Khomyakova.

**Validation:** Irina Amosova, Mikhail Khmelevskii, Andrei Afanasiev, Dmitry Zubkov, Anton Tikhonov, Tao Zhu.

**Visualization:** Tao Zhu.

**Writing – original draft:** Yana Serdyuk, Mikhail Khmelevskii, Nadezhda Khomyakova.

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
