## [Decision Letter · Decision Letter 0]

13 May 2022

PONE-D-21-37203

Immunogenicity and safety of a recombinant adenovirus type-5 COVID-19 vaccine in adults: data from a randomised, double-blind, placebo-controlled, single-dose, phase 3 trial in Russia

PLOS ONE

Dear Dr. Dzutseva,

Thank you for submitting your manuscript to PLOS ONE. After careful consideration, we feel that it has merit but does not fully meet PLOS ONE’s publication criteria as it currently stands. Therefore, we invite you to submit a revised version of the manuscript that addresses the points raised during the review process.

I agree with the reviewers that the manuscript needs attention to details at several places. Particularly, the following must be addressed. First, the construct must be appropriately described as to the length of the transgene. Second, clarity is needed on exclusion criteria, and on the sample sizes used for various analyses. Third, discrepancies in mathematical calculations (percentages) must be addressed. Fourth, figures and panels must be appropriately referred in the text, with appropriate inferences drawn from each of the figures, and it should be made sure that statements are supported by data.

We look forward to receiving your revised manuscript.

Kind regards,

Nagendra R Hegde, B.V.Sc., M.V.Sc., Ph.D.

Academic Editor

PLOS ONE

**Journal requirements:**

“The study was designed, funded, and managed by NPO Petrovax Pharm LLC (Moscow, Russian Federation). NPO Petrovax Pharm LLC in partnership with CanSino Biologics, Inc. (Tianjin, China) are funding and managing the clinical development of the Ad5-nCoV vaccine in the Russian Federation.”

“The study was designed, funded, and managed by NPO Petrovax Pharm LLC (Moscow, Russian Federation). NPO Petrovax Pharm LLC in partnership with CanSino Biologics, Inc. (Tianjin, China) are funding and managing the clinical development of the Ad5-nCoV vaccine in the Russian Federation.”

“All authors have read the journal’s policy and the authors of this manuscript have the following competing interests:. MT, MK, DZ, AA, NK, AT and VD are employees of NPO Petrovax Pharm LLC. TZ and LB are employees of CanSino Biologics, Inc. IA, SS, KZ and YS have received funding from NPO Petrovax Pharm LLC for consultation services. DL and GE have received personal fees from NPO Petrovax Pharm LLC for consultation services.”

4. PLOS requires an ORCID iD for the corresponding author in Editorial Manager on papers submitted after December 6th, 2016. Please ensure that you have an ORCID iD and that it is validated in Editorial Manager. To do this, go to ‘Update my Information’ (in the upper left-hand corner of the main menu), and click on the Fetch/Validate link next to the ORCID field. This will take you to the ORCID site and allow you to create a new iD or authenticate a pre-existing iD in Editorial Manager. Please see the following video for instructions on linking an ORCID iD to your Editorial Manager account: https://www.youtube.com/watch?v=_xcclfuvtxQ.

**Additional Editor Comments:**

*Following consultation with the Academic Editor, we believe that these additional concerns should be addressed to the promote transparency. *

*We have noted that the term ‘superiority’ was used to describe the active vaccine over the placebo. Superiority is when two vaccines are compared in parallel, not when compared to a placebo.  Please revise this accordingly.*

*In the discussion, it is stated that the vaccine produced an antibody response 'similar' to another adenovirus vectored vaccine. However it should be noted that these are two separate studies, not head to head comparisons. *

*It is stated that the vaccine was developed by CanSino and provided by NPO Petrovax. It is not clear whether NPOP produced the vaccine or whether they simply imported the bulk product from CanSino. Please could you clarify this.*

*It is in our understanding that Prometheus is the clinical trial, but it is not clear whether the study is named such or whether Prometheus is the agency that conducted the trial. This requires further clarification since we believe it best practice that a third, independent party was involved in conducting the trial.*

*It is currently not clear form the COI statement that neither CanSino nor NPOP (nor Prometheus?) was involved in the design of the study, data collection or analysis, or interpretation of the results, or  involved in the decision to submit the results for publication. Please could you clarify within the COI statement if this was the case. *

Lucinda Shen

Staff Editor 

on behalf of 

Nagendra R Hegde, B.V.Sc., M.V.Sc., Ph.D.

Academic Editor

PLOS ONE

**Reviewers' comments:**

Reviewer's Responses to Questions

**Comments to the Author**

1. Is the manuscript technically sound, and do the data support the conclusions?

Reviewer #1: Partly

Reviewer #2: Yes

Reviewer #3: Partly

2. Has the statistical analysis been performed appropriately and rigorously? 

Reviewer #1: I Don't Know

Reviewer #2: Yes

Reviewer #3: Yes

3. Have the authors made all data underlying the findings in their manuscript fully available?

Reviewer #1: Yes

Reviewer #2: Yes

Reviewer #3: Yes

4. Is the manuscript presented in an intelligible fashion and written in standard English?

Reviewer #1: No

Reviewer #2: Yes

Reviewer #3: Yes

5. Review Comments to the Author

Reviewer #1: 1. Authors have written observations in the manuscript but explanations of figures are entirely missing.

2. Authors have done statistical analysis but they should explain it a little bit, which can be helpful for a broader audience to understand a clinical study.

4. Manuscript needs major revision. Authors can plan putting their comparisons data in tabular form and then explain them in text.

Reviewer #2: The present study is a multicenter, randomized, double-blind, placebo-controlled, phase 3 trial of a candidate vaccine developed in China and based on Ad5 vector. The study included 500 participants aged 18–85 years (mean age: 41.2 years), the immunogenicity, efficacy and safety of the Ad5-nCoV COVID-19 vaccine was assessed up to 6 months after vaccination. A single dose immunization with the Ad5- nCoV candidate vaccine led to a marked immune response which measured at day 14,28 and 6 months. Antibodies raised against SARS-CoV2 RBD, S and Ad5 antigens with variable ratios. Vaccination showed statistically significant immune response compared with the placebo unvaccinated participants. The study effort is highly appreciated.

We have some comments would that the authors could be reply

1. The sample size was relatively small.

2. The study needs more data about pre-existing immunity to Ad5 between the Russian people to differentiate the neutralizing antibodies raised against the target protein of the candidate SARS-CoV2 vaccine.

3. As a result of virus mutation, the study of immune response to Ad5 cassette for vaccine development is an urgent need.

Some typing corrections :

P.7. Raw 110,211 &229 change urinanalysis to urine analysis

P.9,R.163 correct serums to sera

P. 10,R 166 change reproduction to propagation

P.12, R.219 correct SARS-CoV3 to SARS-CoV2

P.15 paragraph “ Participants were…………………….pandemic” need more clarification

Fig 5 need to use contrasted colors to differentiate between lines

Reviewer #3: The authors present results from a phase III trial of the Ad5-nCoV vaccine versus placebo in approximately 500 individuals in Russia. Specifically, authors focus on results related to immunogenicity, efficacy, reactogenicity and safety. Seroconversion rates against receptor binding protein (RBD), S protein and neutralizing SARS-CoV2 antibodies were high at 28 days post vaccination in the vaccinated group. Geometric mean titres (GMTs) were higher in the vaccinated group than the placebo group though the difference between groups was not as large 6 months post vaccine. Safety was good. The manuscript will be strengthened if the authors consider the following points.

1. To be complete, authors should provide reasons for exclusion for the 283 individuals that were screened but not enrolled - this could be added to Figure 1.

2. The various sample sizes for analyses (at different time points) is a bit confusing. For example, Figure 1 has 120 individuals in the Placebo group for the efficacy analysis but line 326 says there were 119. In line 327, there were 359 in the Ad5-nCoV group for Day 14, but 363 for Day 28 (so did 4 people miss the day 14 assessment?). There also seems to be different numbers of participants in the groups for different outcomes (but same day assessment) - for example, line 352 has 308 individuals for day 14, but for the RBD analyses, there were 359 on day 14. Authors need to clarify the available sample sizes for the different analyses. Also, was drop-out different between the two groups?

3. Authors evaluated cellular immune response on a subgroup of 69 participants. Did these individuals differ in any way from those that did not have this evaluation performed?

4. lines 507-511: no data are presented to support these statements

5. line 528: authors refer to an increase observed at month 6 in the placebo group for several of the GMTs, but I do not think those results were actually presented. The majority of results focused on differences between the vaccine and placebo group.

Minor points:

1. lines 28-30: this sentence is an incomplete sentence. Authors might consider adding "were observed" at the end of the sentence.

2. line 302: Table 1 has 297 females, while the text says there were 297 males. Authors should make the appropriate correction either to the table or the text.

3. Table 1: why were 60% of the participants missing information on underlying disease?

4. In the results section, why are the results for the primary outcome not presented first?

5. line 329: what were the frequency and seroconversion rate in the Placebo group?

6. line 340: change 0/8% to 0.8%

7. line 347: change "significant greater" to "significantly greater"

8. line 351: Authors refer to Figure 3, but I believe this should be Figure 2, since they are presenting GMTs.

9. line 355: authors give a percentage of 59%, but this is not the same as 183/320, so something should be corrected there.

10. Authors should refer to Figure 3 after presenting seroconversion rates.

11. line 356: authors say there were differences on day 14, but this is not noted in Figure 3.

12. Figure 3 caption: there is no Day 0 presented in this figure.

13. Figure 7 - the median and quartiles are difficult to see in the figure. Authors should try to make this more clear.

14. lines 450-455: authors should make clear which group the percentages corresponds to...as it is now, it is confusing, for example, to see 13/31 [11.32%] since 13/31 is not 11.32%.

15. line 458: Table 2 presents safety information for all participants, not just those receiving the Ad5-nCoV vaccine. (113 out of 496 reported systemic reactions...if authors want to focus on the Ad5-nCoV group, it would be 100 out of 372)

16. In S1 Table the number with at least one AE in the Placebo group should be higher than 2 (13 maybe?). The percentage for the Placebo group and for the total group should be corrected.

17. lines 468-470: this sentence should be rephrased, since there were a total of 5 severe reactions, which included 4 with an increase in body temperature and 1 with myalgia.

18. S2 Table: The numbers for Grade 1 and Grade 2 body temperature increases are confusing - for example, what is meant by 58/64?

19. line 472 - I believe the percentage for fatigue is incorrect (it doesn't match Table S2).

20. line 484 - make clear that the "most common reactions" refers to the vaccine group.

21. line 484 - the percentage for induration does not match what is presented in Table S4.

22. line 486 - given the rest of the sentence, why is this percentage out of all of the participants in the group?

23. line 491: what does 38/75 refer to (specifically the 75?), since there are more than 75 people in the Placebo group.

24. lines 498-499 - authors should cross-check the percentages in the text with those presented in Table S6, since there are some differences.

6. PLOS authors have the option to publish the peer review history of their article (what does this mean?). If published, this will include your full peer review and any attached files.

Reviewer #1: No

Reviewer #2: No

Reviewer #3: No

---

## [Author Response · Author response to Decision Letter 0]

25 Jul 2022

Dear Editor Nagendra R. Hegde,

Thank you for giving us the opportunity to submit a revised manuscript, titled “Immunogenicity and safety of a recombinant adenovirus type-5 COVID 19 vaccine in adults: data from a randomised, double-blind, placebo-controlled, single-dose, phase 3 trial in Russia” for publication in PLOS One. We appreciate the time and effort that you and the reviewers have dedicated to providing your valuable feedback on this manuscript. We have been able to incorporate changes to reflect the suggestions provided by the reviewers. For your and reviewers’ convenience, all changes are marked using track changes in the revised manuscript, that is, in the ‘Revised Manuscript with Track Changes.docx’ file. 

Please find a point-by-point response to the reviewers’ comments and concerns below here. Please find the black comments followed by our reply in red.

As requested, our updated competing interest statement is as follows: All authors have read the journal’s policy and the authors of this manuscript have the following competing interests: MT, MK, DZ, AA, NK, AT and VD are employees of NPO Petrovax Pharm LLC. TZ and LB are employees of CanSino Biologics, Inc. IA, SS, KZ and YS have received funding from NPO Petrovax Pharm LLC for consultation services. DL and GE have received personal fees from NPO Petrovax Pharm LLC for consultation services. This does not alter our adherence to PLOS ONE policies on sharing data and materials.

We are looking forward to your response.

On behalf of the study authors, 

Vitalina Dzutseva

Reviewer 1

Comment 1: Authors have written observations in the manuscript but explanantions of figures are entirely missing.

Authors’ Response: Through the insightful comments provided by the reviewers and editors, we have thoroughly revised the manuscript. As part of these revisions, we have added more detail on the experiments themselves, results, and statistical analysis performed to the Results section (pgs. 15–29) and each Figure Legend in the revised manuscript. Combined, these revisions should help clarify our interpretation and viewpoint of the data to the reader.

Comment 2: Authors have done statistical analysis but they should it a little bit, which can be helpful for a broader audience to understand a clinical study.

Authors’ Response: We have condensed the description of the Statistical Methods in the main body of the manuscript (pgs. 13–14, lines 256–274). We have also moved the section describing the participant populations to the section on Randomisation in the Methods section (pgs. 8–9, lines 138–151). For transparency and to promote analytical reproducibility, we have also retained the original description of the statistical methods as supporting information (S1 Methods). The S1 Methods also contains further details on the selection of sample size, such as the smaller cohort used for the cellular immune response analysis.

Comment 3: Manuscript needs major revision. Authors can plan putting their comparisons in tabular form and then explain them in the text.

Authors’ Response: Through the insightful comments from the reviewers and editors, the manuscript has undergone major revisions. These revisions include, but certainly are not limited to, revising the description of the Figures in the Results section (pgs. 15–26). In addition, we have generated a table (S1 Table) to complement their graphical representation (Figs. 2 and 3). We have also added another table to present the rate of IFN-�-positivity among participants (Table 2; pg. 25, lines 499–507). 

Reviewer 2

Comments:

The present study is a multicenter, randomized, double-blind, placebo-controlled, phase 3 trial of a candidate vaccine developed in China and based on Ad5 vector. The study included 500 participants aged 18–85 years (mean age: 41.2 years), the immunogenicity, efficacy and safety of the Ad5-nCoV COVID-19 vaccine was assessed up to 6 months after vaccination. A single dose immunization with the Ad5- nCoV candidate vaccine led to a marked immune response which measured at day 14,28 and 6 months. Antibodies raised against SARS-CoV2 RBD, S and Ad5 antigens with variable ratios. Vaccination showed statistically significant immune response compared with the placebo unvaccinated participants. The study effort is highly appreciated.

We have some comments would that the authors could be reply:

Comment 1: The sample size was relatively small.

Authors’ Response: Based on previous studies (as described in the Methods section), the power calculation performed, allowing for a 90% chance of detecting a statistical difference in the primary outcome measure and a 10% drop-out rate, required a study cohort of 200 participants. However, we shared the reviewer’s concerns regarding the sample size when designing the trial. We therefore increased this number to 500, thereby expanding the breadth of the data accrued on safety and efficacy. This was an ethical consideration that was taken to moderate the overall burden of the participants in this study. Moreover, in accordance with Russian regulations that categorized this study as a Phase 3 immunogenicity study, the sample size was comparable to numbers used in similar trials.1–3

As another consideration, while conducting the trial between 11-September-2020 to 05-May-2021, the number of new COVID-19 cases peaked to approximately 30,000 new cases per day in December 2020 in Russia.4 At this stage of the pandemic, it was challenging to recruit additional healthy volunteers that met the inclusion criteria; moreover, some participants received the publically available vaccine when it became available in December 20205,6 and did not report it, despite our best efforts, which may have influenced some data as we discussed in the manuscript (pg. 31, lines 608–611). 

In the separate cohort that was used to evaluate vaccine efficacy (i.e., the cellular immune response), a sample size of 60 participants was selected. This cohort was a subset of participants from the FAS for immunogenicity analysis population. The selection of at least 60 participants was not based on a calculation, but rather logistics. The cellular immune response analysis was performed at the Moscow clinic site and 69 participants were the number of participants visiting that particular site during the trial. The locations of the other clinic sites in Russia were too far away to collect the samples and perform the analysis all within 24 hours. Therefore, only participants that visited the Moscow site were used for this specific analysis. Moreover, as the similar phase 2 clinical study in China used 32 participants for the same type of analysis,3 we were confident that we would observe meaningful results. 

To emphasize the suitability of the sample size, we have revised the text in the S1 Methods section as follows (pg. 1, lines 13–17):

References:

1. Ella R, Reddy S, Jogdand H, Sarangi V, Ganneru B, Prasad S, et al. Safety and immunogenicity of an inactivated SARS-CoV-2 vaccine, BBV152: interim results from a double-blind, randomised, multicentre, phase 2 trial, and 3-month follow-up of a double-blind, randomised phase 1 trial. Lancet Infect Dis. 2021;21(7):950-961. doi: 10.1016/S1473-3099(21)00070-0.

2. Ramasamy MN, Minassian AM, Ewer KJ, Flaxman AL, Folegatti PM, Owens DR, et al. Safety and immunogenicity of ChAdOx1 nCoV-19 vaccine administered in a prime-boost regimen in young and old adults (COV002): a single-blind, randomised, controlled, phase 2/3 trial. Lancet. 2021;396(10267):1979-1993. doi: 10.1016/S0140-6736(20)32466-1.

3. Zhu F, Guan X, Li Y, Huang J, Jiang T, Hou L, et al. Immunogenicity and safety of a recombinant adenovirus type-5-vectored COVID-19 vaccine in healthy adults aged 18 years or older: a randomised, double-blind, placebo-controlled, phase 2 trial. Lancet. 2020;396: 479–488. doi: 10.1016/S0140-6736(20)31605-6.

4. Statista.com. ”Number of new daily coronavirus (COVID-19) cases confirmed in Russia from January 31, 2020 to June 10, 2022.” Accessed from: https://www.statista.com/statistics/1102303/coronavirus-new-cases-development-russia/

5. Logunov DY, Dolzhikova IV, Shcheblyakov DV, Tukhvatulin AI, Zubkova OV, Dzharullaeva AS, et al. Safety and efficacy of an rAd26 and rAd5 vector-based heterologous prime-boost COVID-19 vaccine: an interim analysis of a randomised controlled phase 3 trial in Russia. Lancet. 2021;397(10275):671-681. doi: 10.1016/S0140-6736(21)00234-8. 

6. Baraniuk C. Covid-19: What do we know about Sputnik V and other Russian vaccines? BMJ. 2021;372:n743. doi: 10.1136/bmj.n743.

Comment 2: The study needs more data about pre-existing immunity to Ad5 between the Russian people to differentiate the neutralizing antibodies raised against the target protein of the candidate SARS-CoV2 vaccine. 

Authors’ Response: We agree that pre-existing immunity to Ad5 may affect the efficacy of the immune response against the vaccine. It was not possible to determine pre-existing immunity amongst the study cohort against Ad5. However, it is clear that the S protein from SARS-CoV-2 still elicited an adequate immune response, as determined by the production of neutralizing antibodies. These data suggest that any pre-existing immunity against Ad5 did not impact the efficacy of the vaccine. 

We agree with the reviewer that the pre-existing immunity to Ad5 is an avenue to be further explored. As referenced in our response to the previous comment, some participants received the publically available vaccine, which may have inadvertently affected the results as discussed in the manuscript (pg. 31, lines 608–611). In the future, especially now that the height of the pandemic has past, we plan to explore this research area further in a larger cohort.

Comment 3: As a result of virus mutation, the study of immune response to Ad5 cassette for vaccine development is an urgent need.

Authors’ Response: We agree with the reviewer and thank them for this perceptive comment. We hope to address this point in future studies. We have added the following sentence to the Discussion (pg. 32, lines 636–637):

“Studies with longer follow-up periods would help to clarify the longevity of the immune response in those vaccinated with Ad5-nCoV, particularly with multiple COVID-19 variants circulating amongst the global population.”

Comment 4: Some typing corrections: 

P.7. Raw 110,211 &229 change urinanalysis to urine analysis

P.9,R.163 correct serums to sera

P. 10,R 166 change reproduction to propagation

P.12, R.219 correct SARS-CoV3 to SARS-CoV2

Authors’ Response: Thank you for pointing out these errors. We have implemented your suggestions in the revised manuscript.

Comment 5: P.15 paragraph “ Participants were…………………….pandemic” need more clarification

Authors’ Response: The original text stated, ”Participants were not directly involved in the development, implementation, or interpretation of this study due to the requirement for a quick response to the rapidly-evolving coronavirus pandemic.” 

To clarify the text, we have revised the text to the following (pg. 14, lines 276–277): 

“Participants were fully informed about the study and informed consent was obtained. We thank them for their involvement in the study.”

Comment 6: Fig 5 need to use contrasted colors to differentiate between lines

Authors’ Response: Based on your recommendation, we have changed the colour scheme to rainbow, thereby making it easier to distinguish between the different GMT threshold levels. The revised figure is shown below (Figure 5).

Reviewer 3 

The authors present results from a phase III trial of the Ad5-nCoV vaccine versus placebo in approximately 500 individuals in Russia. Specifically, authors focus on results related to immunogenicity, efficacy, reactogenicity and safety. Seroconversion rates against receptor binding protein (RBD), S protein and neutralizing SARS-CoV2 antibodies were high at 28 days post vaccination in the vaccinated group. Geometric mean titres (GMTs) were higher in the vaccinated group than the placebo group though the difference between groups was not as large 6 months post vaccine. Safety was good. The manuscript will be strengthened if the authors consider the following points.

Major Comments

Comment 1: To be complete, authors should provide reasons for exclusion for the 283 individuals that were screened but not enrolled - this could be added to Figure 1.

Authors’ Response: These 283 participants did not meet the eligibility criteria, and therefore were screening failures. We have amended Figure 1 to include this information. The revised figure (Fig. 1) is shown below.

Fig. 1

Comment 2: The various sample sizes for analyses (at different time points) is a bit confusing. For example, Figure 1 has 120 individuals in the Placebo group for the efficacy analysis but line 326 says there were 119. In line 327, there were 359 in the Ad5-nCoV group for Day 14, but 363 for Day 28 (so did 4 people miss the day 14 assessment?). There also seems to be different numbers of participants in the groups for different outcomes (but same day assessment) - for example, line 352 has 308 individuals for day 14, but for the RBD analyses, there were 359 on day 14. Authors need to clarify the available sample sizes for the different analyses. Also, was drop-out different between the two groups?

Authors’ Response: We apologize for the confusion in the text. The numbers mentioned in this section of the text referenced the total number of participants with complete titre data that was used perform the seroconversion analysis, and at some days, not all participants were available or data points were missing owing to protocol deviations at some sites (i.e., some were not performed), which is why there is some variations between the days. As noted in the Protocol (S1 Protocol), Visits 4 and 5 were to perform safety and immunogenicity evaluations on Days 14 and 28, respectively. As you have pointed out on Day 14 in the Ad5-nCoV group, there were 4 participants that missed Day 14. These deviations that occurred in the Ad5-nCoV and Placebo groups are noted in Figure 1.

Following your comment, we also acknowledge that presentation of the study populations could be revised for clarity. As a result, we have revisited the populations used for analysing the primary and secondary endpoints in this study. Owing to the similarities between the (full analysis set) FAS and (per-protocol set) PPS for immunogenicity analysis, we have used the FAS for immunogenicity analysis throughout the study. Clarification on the study populations are provided in the methods (pgs. 8–9, lines 138–151), Fig. 1 itself and its accompanying figure legend, and the description of Fig. 1 in the results (pg. 15, lines 284–291; pg. 16, lines 310–313). The revised Fig. 1 is shown in our response to Comment 1.

After specifying the populations used, to clarify the text and present the data in a more concise manner, we have generated a table (S1 Table in the revised manuscript) that lists the total number of participants for the Placebo and Ad5-nCoV groups, the total number of participants with seroconversion data available, and the number of participants who seroconverted. In addition, we have specified the populations used in the analysis presented by each figure and table in its accompanying legend and caption/footnote, respectively. 

Additionally, we have defined the seroconversion rate in the text as follows (pg. 19, lines 350–352):

“Seroconversion rates, that is, the proportion of participants (out of total participants with seroconversion data) with at least four-fold increase in antibody titres after vaccination, were examined (Fig. 3 and Table S1).”

Comment 3: Authors evaluated cellular immune response on a subgroup of 69 participants. Did these individuals differ in any way from those that did not have this evaluation performed?

Authors’ Response: As mentioned in response to Comment 1 from Review 2, a study cohort of 60 participants was used to evaluate vaccine efficacy and this subset was derived from the FAS for immunogenicity analysis population. No other differences were present between the participants recruited for this cohort compared to the larger cohort used from the beginning of the study.

Part of the rationale behind the sample size was attributed to the use of only the Moscow site (National Research Center for Hematology, Moscow) as it was the site best equipped to perform the analysis. Additionally, the collected blood samples would need to be transported and analysed within 24 hours of collection while maintaining a liquid state. Therefore, the other Russian clinical sites were not suitable as the distances between the sites to the Moscow clinic were quite large. Additionally, the phase 2 study from China that this study was based on used 32 participants for the same type of analysis.3

We have clarified the text by introducing the revised text in S1 Methods section as discussed in response to Comment 1 from Reviewer 2 (pg. 1, lines 13–17).

Comment 4: lines 507-511: no data are presented to support these statements

Lines in question: Results of the analysis of laboratory parameters demonstrated a trend towards an increase in reactive protein levels, an increase in the red cell sedimentation rate, an increase in the mean percentage of monocytes, and a decrease in the mean percentage of neutrophils after administration of Ad5-nCoV. The changes in the examined laboratory parameters were pronounced the day after vaccination, but had largely resolved by Day 28.

Authors’ Response: These data refer to our laboratory adverse events, none of which were severe or serious. These laboratory adverse events are summarised in an earlier paragraph in the results section and reference the data shown in S7 Table (in the revised manuscript). To emphasize our findings, we have added another reference to S7 Table after the list of laboratory adverse events (pg. 29, line 577).

Comment 5: line 528: authors refer to an increase observed at month 6 in the placebo group for several of the GMTs, but I do not think those results were actually presented. The majority of results focused on differences between the vaccine and placebo group.

Authors’ Response: These results were presented in Figure 2 in which the GMT for the respective antibodies examined (particularly anti-RBD and anti-S protein) were elevated at 6 months post-vaccination. We have revised the text and Figure panels to help clearly convey our meaning. The revised text in the Results section is as follows (pg. 18, lines 328–348):

” Participants that received the Ad5-nCoV vaccine exhibited a significant immunogenic response against the S protein, with antibody levels significantly increased at Day 14, Day 28 and Month 6 compared with the placebo (p<0.001; Fig. 2A and S1 Table). Anti-RBD levels were significantly higher at Day 14 and at Day 28 compared with placebo (p<0.001), but after 6 months, GMTs had waned and dropped to levels consistent with those observed in the Placebo group by Month 6 (Fig. 2B and S1 Table). Interestingly, a notable increase in anti-S protein and anti-RBD antibodies at Month 6 was also observed in the Placebo group (Fig. 2A and 2B and S1 Table), which could be attributed to asymptomatic infections.

Next we examined if the antibodies offered protein against SARS-CoV-2 infection. We found that NAbs were significantly increased by Day 14 in the Ad5-nCoV group compared with levels in the Placebo group, and levels remained significantly higher at 6 Month (p<0.001; Fig. 2C and S1 Table). We did observed an approximate two-fold increase in antibody titre levels in the Placebo group by Month 6, which may be indicative of community transmission of SARS-CoV-2. Participants that received the vaccine also exhibited a significant rise in Ad5 antibody levels compared with the Placebo group at Day 28 and at Month 6 (p<0.001; Fig. 2D and S1 Table). Notably, NAbs and Ad5 antibodies were also elevated at Month 6 in the Placebo group (Fig. 2C and 2D and S1 Table), which could again be attributed to asymptomatic infections or if participants did not report receiving the publically available vaccine that was available in December 2020 [17,18]. Taken together, these data suggest that the Ad5-nCoV vaccine elicits a strong immunogenic response to the S protein, specifically against RBD, with immunity to SARS-CoV-2 infection lasting 6 months.

.”

Minor comments

Comment 1: lines 28-30: this sentence is an incomplete sentence. Authors might consider adding "were observed" at the end of the sentence.

Authors’ Response: Thank you for bringing this to our attention. We have incorporated your suggestion into the revised manuscript.

Comment 2: line 302: Table 1 has 297 females, while the text says there were 297 males. Authors should make the appropriate correction either to the table or the text.

Authors’ Response: Thank you for bringing this to our attention. We have corrected this oversight in the revised manuscript.

Comment 3: Table 1: why were 60% of the participants missing information on underlying disease?

Authors’ Response: We have reviewed our statistical report and noted inconsistencies in the presentation of the data. Numbers may have been attributed to our interim analysis and the table was not fully updated. We apologise for this oversight. The data has been reviewed and verified in Table 1 (as shown below; changes are in red).

Table 1. Participant Demographics (Safety Analysis Set).

 Ad5-nCoV

N=372 Placebo

N=124 Total

N=496

Age (years[%]) 

18–44 years 223 (59.9) 77 (62.1) 300 (60.5)

45–59 years 122 (32.8) 39 (31.5) 161(32.5)

≥60 years 27 (7.3) 8 (6.5) 35 (7.1)

Mean 41.2 41.0 41.2

Sex (n[%]) 

Male 151 (40.6) 48 (38.7) 199 (40.1)

Female 221 (59.4) 76 (61.3) 297 (59.9)

Race (n[%]) 

White 371 (99.7) 122 (98.4) 493 (99.4)

Asian 1 (0.3) 2 (1.6) 3 (0.6)

Country (n[%]) 

 Russia 372 (100%) 124 (100%) 496 (100%)

Body Mass Index (kg/m2) 

Mean 25.03 24.7 24.94

Minimum 18.6 18.6 18.5 

Maximum 30.0 29.9 30.0

Underlying Disease (n[%]) 

Yes 98 (26.3) 29 (23.4) 127 (25.6)

No 274 (73.7) 95 (76.6) 369 (74.4)

Prior Disease (n[%]) 

Yes 176 (47.3) 56 (45.2) 232 (46.8)

No 196 (52.7) 68 (54.8) 264 (54.2)

Comment 4: In the results section, why are the results for the primary outcome not presented first?

Authors’ Response: Geometric mean titres are used to calculate the seroconversion rate, which is defined as a four-fold increase in antibody titres. As GMTs are necessary to help calculate the seroconversion rates, and thus evaluating the primary outcome, we have maintained the current order of data. To emphasize that seroconversion is the primary outcome and that determining titre levels are required, we have added the definition for seroconversion as discussed in our response to Comment 2 from Reviewer 3.

Comment 5: line 329: what were the frequency and seroconversion rate in the Placebo group?

Authors’ Response: We have compiled all of this data to generate a new table, S1 Table in the revised manuscript. The frequency and seroconversion rates for all conditions are provided.

Comment 6: line 340: change 0/8% to 0.8%

Authors’ Response: Thank you for bringing this to our attention. We have corrected this oversight in the revised manuscript.

Comment 7: line 347: change "significant greater" to "significantly greater"

Authors’ Response: Thank you for bringing this to our attention. We had incorporated your suggestion, but owing to the extensive revisions of the manuscript, this exact text and change is no longer present.

Comment 8: line 351: Authors refer to Figure 3, but I believe this should be Figure 2, since they are presenting GMTs.

Authors’ Response: Initially when writing the results, we attempted to explain the data by antibody being examined (RBD, S protein, etc.), but we acknowledge that this has led to confusion. Accordingly, we have markedly revised the results section so that the Figures are discussed in the same order as they are presented as discussed in our response on Figure 2 in Comment 5. The results for Figures 3 and 4 have also followed this revised format (pgs. 19–21) 

Comment 9: line 355: authors give a percentage of 59%, but this is not the same as 183/320, so something should be corrected there.

Authors’ Response: We have checked the data and noted a typo, the numbers should be 183/310, which yields 59%. The data has been reviewed, corrected if required and compiled into S1 Table (in the revised manuscript). 

Comment 10: Authors should refer to Figure 3 after presenting seroconversion rates.

Authors’ Response: Figure 3 itself presents the seroconversion rates. We have revised the text and added S1 Table, which also presents the seroconversion rates as well as the seroconversion frequency within the participant cohorts for participants with seroconversion data. The revised text for Figure 3 is described on pgs. 19–20, lines 350–382.

Comment 11: line 356: authors say there were differences on day 14, but this is not noted in Figure 3.

Authors’ Response: Thank you for raising this point as the significance was not included in Figure 3 as it should have been. We have revised Figure 3 to include these symbols at Day 14 for neutralising antibodies against SARS-CoV-2 in the Ad5-nCoV group. The text describing this aspect of Figure 3 is as follows (pg. 19, lines 352–356):

“For those vaccinated with Ad5-nCoV vs. the placebo, seroconversion rates for S protein (90.6% [95% CI: 87.2, 93.4] vs. 6.72% [95% CI: 2.95, 12.8]), RBD (78.5% [95% CI: 73.9, 82.6] vs. 5.88% [95% CI: 2.4, 11.7]), and NAbs (78.5% [95% CI: 73.9, 82.6] vs. 3.88% [95% CI: 1.07, 9.65]) were significantly elevated at Day 14 and peaked on Day 28 (p<0.001; Fig. 3A–3C and S1 Table).”

Figure 3

Comment 12: Figure 3 caption: there is no Day 0 presented in this figure.

Authors’ Response: This is correct—no Day 0 should be present as the seroconversion rates are calculated based on the fold increase from baseline (i.e., Day 0). In the S1 Table that accompanies Figure 3, Day 0 points have not been included to indicate that there is not calculation to input. By also adding the seroconversion rate definition to the text and as a footnote below S1 Table, this should also help clarify why there is no data for Day 0. Please see our responses to Comment 11 that highlight the text changes in the revised manuscript.

Comment 13: Figure 7 - the median and quartiles are difficult to see in the figure. Authors should try to make this more clear.

Authors’ Response: We have revised Figure 7 by making the data points more transparent, this has helped to improve visibility of the median and quartiles. Revised Figure 7 is shown below: 

Figure 7

Comment 14: lines 450-455: authors should make clear which group the percentages corresponds to...as it is now, it is confusing, for example, to see 13/31 [11.32%] since 13/31 is not 11.32%.

Authors’ Response: We were referencing the number of participants with COVID from the Placebo group (n=13) and Ad5-nCoV group (n=18). For clarity, we have revised the text and use the previously defined per-protocol Set (Fig. 1). The revised text in the Results section states as follows (pg. 17, lines 304–307):

“Within the PPS for efficacy analysis population, 13 cases of COVID-19 occurred in participants from the Placebo group (13/120 [10.83%]), whereas 18 cases of COVID-19 occurred in participants from the Ad5-nCoV group (18/361[4.99%]; p=0.024).”

Comment 15: line 458: Table 2 presents safety information for all participants, not just those receiving the Ad5-nCoV vaccine. (113 out of 496 reported systemic reactions...if authors want to focus on the Ad5-nCoV group, it would be 100 out of 372)

Authors’ Response: The reviewer is correct that we wish to focus on those in the Ad5-nCoV group. We have modified the text to reflect the numbers for the Ad5-nCoV group from Table 3 (pg. 26; line 523).

Comment 16: In S1 Table the number with at least one AE in the Placebo group should be higher than 2 (13 maybe?). The percentage for the Placebo group and for the total group should be corrected.

Authors’ Response: Thirteen is correct, thank you for pointing this out. We can confirm that systemic reactions were reported in 13 (10.5%) participants from the Placebo group. We have modified the Table (Table 3 in the revised manuscript). The associated text that described this table in the Abstract (pg. 2, line 38) and Results sections (pg. 26, line 525) had the correct information. 

Comment 17: lines 468-470: this sentence should be rephrased, since there were a total of 5 severe reactions, which included 4 with an increase in body temperature and 1 with myalgia.

Authors’ Response: Thank you for bringing this to our attention. We have incorporated your suggestion into the revised manuscript (pg. 27, lines 535–537).

Comment 18: S2 Table: The numbers for Grade 1 and Grade 2 body temperature increases are confusing - for example, what is meant by 58/64?

Authors’ Response: We apologize for the confusion. We have inadvertently left out an important footnote for this table. For Grades 1 and 2 body temperature increases, the numbers represent the number of participant/number of AEs. This is because more a participant experienced several cases of the reported AE (body temperature increase) during the study. To clarify the text and table, we have added the following footnote to the table (S3 Table in the revised manuscript):

“a For the AE at the indicated severity, n represents the number of participants/number of AEs as one participant experienced several cases of the same AE during the study.”

Comment 19: line 472 - I believe the percentage for fatigue is incorrect (it doesn't match Table S2).

Authors’ Response: We acknowledge the error in line 472 and that the percentage for fatigue should be 4.8%. In the revised manuscript, we have corrected the text (pg. 28, line 539). 

Comment 20: line 484 - make clear that the "most common reactions" refers to the vaccine group.

Authors’ Response: Thank you for bringing this to our attention. We have incorporated your suggestion into the revised manuscript (pg. 28; line 551).

Comment 21: line 484 - the percentage for induration does not match what is presented in Table S4.

Authors’ Response: Thank you for bringing this to our attention. We have incorporated your suggestion into the revised manuscript (pg. 28; line 552).

Comment 22: line 486 - given the rest of the sentence, why is this percentage out of all of the participants in the group?

Authors’ Response: The original sentence stated: “Fifteen participants (4.0%) who received the Ad5-nCoV vaccine had a moderate reaction that included pain, erythema, swelling or induration at the injection site.”

A total number was provided for all of the moderate (Grade 2) reactions to provide a concise scope for all of the local post-vaccination reactions experienced while also emphasizing that these reactions occurred rarely (i.e., in 4% of participants). To maintain a similar format, we have revised the sentence to the following (pg. 28, lines 553–556):

While moderate reactions rarely occurred in those that received the Ad5-nCoV vaccine, 15 participants (4%) in the Ad5-nCoV group experienced the following local post-vaccination reactions at the injection site: erythema (8 [1.6%]), pain (6 [1.2%]), swelling (5 [1.0%]), and induration (1 [0.2%]).

Comment 23: line 491: what does 38/75 refer to (specifically the 75?), since there are more than 75 people in the Placebo group.

Authors’ Response: As mentioned in Comment 18, n in this case represents the number of participants/number of AEs as a participant could experience several cases of the same AE during the study. Similar to the footnote mentioned in Comment 18, we have amended the note that is below Table 3 for clarity (pg. 27, lines 528–529). Additionally, the numbers in Table 3 have been rearranged to match the description of the footnote. The revised note below Table 3 states:

“For AEs at the indicated severity with two numbers, n represents the number of participants per category/number of AEs as one participant experienced several cases of the same AE during the study.”

Comment 24: lines 498-499 - authors should cross-check the percentages in the text with those presented in Table S6, since there are some differences.

Authors’ Response: Thank you for this recommendation. We have revised the text accordingly to ensure that numbers correspond to the information presented in S7 Table (pg. 29, lines 566–567).

 

Editor’s Comments

The authors present a study based on the phase 3 clinical data available for Ad5-nCoV in the European population. The study revolves about vaccinating a population with a single dose of Ad5-nCoV and others with placebo for the phase 3 trials. I have few suggestions so that the work presented here becomes more interesting for a broader audience.

Comment 1: I will strongly suggest revising the manuscript by compiling their data into a tabular form rather than writing similar sentences throughout the manuscript. Look at sentences: line number 325-358 and similar sentences. 

Authors’ Response: In accordance with yours and Reviewer 1’s suggestion, we have created tables to also illustrate the data graphically represented in Figures 2 and 3. These tables are located in the supporting information, S2 and S3 Tables, respectively, in the revised manuscript.

Comment 2: In Abstract, line number 30: I guess that the authors are trying to say “seroconversion against the SARS-CoV-2 virus” not “neutralizing SARS-CoV-2 antibodies”.

Authors’ Response: We have revised the text to clarify our intended meaning based on your suggestion. The text in the Abstract now states (pg. 2, lines 28–31):

“Seroconversion (the primary endpoint) rates of 78.5% (95% CI: 73.9; 82.6) against receptor binding domain (RBD), 90.6% (95% CI: 87.2; 93.4) against S protein and 59% (95% CI: 53.3; 64.6) seroconversion of neutralising antibodies against SARS-CoV-2 at 28 days post-vaccination were observed.”

Comment 3: Line 34 and 169: Authors should check if full length Spike protein was used in their study or the ectodomain of the spike protein. The spike protein of the SARS-CoV-2 is a type-I transmembrane protein; therefore, it would require the presence of detergents or other agents, which can keep them stable and soluble. 

Authors’ Response: As mentioned in the Methods section (pg. 8, line 171), the “…S protein (amino acids 1–1213)…” was used, which, as the Reviewer has implied, refers to the ectodomain of the S protein. To distinguish between the RBD fragment and other reagents in the paper, we referenced the full length S protein instead of the truncated ectodomain, which we acknowledge may have been unintentionally misleading. 

To avoid confusion, we have revised the text in the Methods to specify that “…SARS-CoV-2 S protein (amino acids 1–1213), a truncated variant that contains the ectodomain of S protein (i.e., recombinant S protein ectodomain)...” (pg. 10, lines 187–188). Accordingly, we instead specified ”recombinant S protein ectodomain” throughout the text. 

Comment 4: Line 151: It should be anti-SARS-CoV-2 virus not “anti-coronavirus”. Both have different meanings.

Authors’ Response: Thank you for pointing out this error. We have implemented your suggestion in the revised manuscript.

Comment 5: Line 543: Authors should provide a citation and try to reframe the sentence. 

Authors’ Response: The original sentence stated, “Our data analysis used a cut-off of 1:200 in line with the Chinese studies.” To clarify this sentence, we have revised the sentence and add the appropriate references, which have been used previously in the manuscript (No. 9: Zhu FC, et al. Lancet 2020; and No. 10: Zhu F, et al. Lancet 2020). The sentence now states (pg. X, line x):

”Our data analysis used a cut-off of 1:200 in line with the previous Chinese studies [9, 10]...”

Comment 6: In the result section: I find that an explanation of their findings and observations are totally missing. They should clearly write: what is the take home message from every experiment. They have written observations but the scientific explanations are missing in their manuscript.

They should also clearly point out in their text, which figure they are discussing. For example, if Fig2 or Fig3 contains multiple figures then they should be marked as A, B, C, and D, and then each of them should be discussed in the text. Otherwise, it is difficult for a reader to grasp all the information. 

Authors’ Response: As recommended by you and the other Reviewers, we have revised the Results section to improve the flow and presentation of data. We have also created a new table (S1 Table in the revised manuscript) to help convey all the important data points based on Figures 2 and 3. Additionally, we have labelled the subpanels as A, B, C, and D as well as rearranged their order to help improve the flow of data and resulting discussions. Changes to the text in the Results are located on pgs. 17–19.

Comment 7: Overall, the manuscript needs a major revision from authors so that it becomes more informative and interesting.

Authors’ Response: In combination with the helpful suggestions provided by the reviewers and editors, the manuscript has been thoroughly revised. 

 

Additional Editor’s Comments

Comment 1: We have noted that the term ’superiority’ was used to describe the active vaccine over the placebo. Superiority is when two vaccines are compared in parallel, not when compared to a placebo. Please revise this accordingly.

Authors’ Response: Thank you for bringing this distinction to our attention. We have replaced superior/superiority in lines 36, 245 and 526 in the revised manuscript.

Comment 2: In the discussion, it is stated that the vaccine produced an antibody response ’similiar’ to another adenovirus vectored vaccine. However, it should be noted that these are two separate studies, not head to head comparisons.

Authors’ Response: Thank you for bringing this issue to our attention.

The original text stated, ”The Ad5-nCoV vaccine provided a similar antibody response to Ad26.COV2.S: 95.9% of subjects developed antibodies to the S protein and 92.5% to the smaller anti-RBD region by Day 28.”

We have revised the sentence as follows (pg. 33, lines 654–656): ”In a previous study, the Ad5-nCoV vaccine provided a similar antibody response as Ad26.COV2.S in which 95.9% of subjects developed antibodies to the S protein and 92.5% to the smaller anti-RBD region by Day 28 [24].”

Comment 3: It is stated that the vaccine was developed by CanSino and provided by NPO Petrovax. It is not clear whether NPOP produced the vaccine or whether they simply imported the bulk product from CanSino. Please could you clarify this.

Authors’ Response: Apologies that this information was not clearly conveyed in the original text. The vaccine and placebo drug products were produced by Cansino. Secondary packaging and quality control was performed by Petrovax. To clarify our meaning, we have added the following text to the Materials and Methods section (pg. 6, lines 81–83):

“The Ad5-nCoV vaccine and placebo drug products were produced by CanSino Biologics Inc [8]. Secondary packaging and quality control were performed by NPO Petrovax.”

Comment 4: It is in our understanding that Prometheus is the clinical trial, but it is not clear whether the study is named such or whether Prometheus is the agency that conducted the trial. This requires further clariufication since we believe it best practice that a third, independent party was involved in conductin the trial.

Authors’ Response: Promethus Rus is the clinical protocol code that was approved by a local authority. We have clarified the text by adding the following sentence (pg. 5; lines 69–71):

”A collaborative development project between CanSino Biologics and the Russian pharmaceutical company NPO Petrovax Pharm LLC provided the basis for a new phase 3 study, Prometheus Rus, which is the clinical protocol code that was approved by a local authority.”

Comment 5: It is currently not clear form the COI statement that neither CanSino nor NPOP (nor Prometheus?) was involved in the design of the study, data collection or analysis, or interpretation of the results, or involved in the decision to submit the results for publication. Please could you clarifdy within the COI statement if this was the case.

Authors’ Response: Petrovax and CanSino are co-sponsors of this study. They were involved in the design of the study, data collection and analysis, and interpretation of results and the decision to submit. We have updated the the COI in the statement to reflect their roles. In the revised text, the COI statement now states the following (pg. 36, lines 719–724):

“All authors have read the journal’s policy and the authors of this manuscript have the following competing interests: MT, MK, DZ, AA, NK, AT and VD are employees of NPO Petrovax Pharm LLC. TZ and LB are employees of CanSino Biologics, Inc. IA, SS, KZ and YS have received funding from NPO Petrovax Pharm LLC for consultation services. DL and GE have received personal fees from NPO Petrovax Pharm LLC for consultation services. This does not alter our adherence to PLOS ONE policies on sharing data and materials.”

Additionally, we have revised the text surrounding the funding and role of the study sponsor (pg. 36; lines 712–717): 

“NPO Petrovax Pharm LLC (Moscow, Russian Federation) and CanSino Biologics, Inc. (Tianjin, China) were co-sponsors of this study. The study was funded and managed by NPO Petrovax Pharm LLC. NPO Petrovax Pharm LLC and CanSino Biologics, Inc. were involved in the design of the study, data collection and analysis, interpretation of results, and the decision to submit. NPO Petrovax Pharm LLC in partnership with CanSino Biologics, Inc. are funding and managing the clinical development of the Ad5-nCoV vaccine in the Russian Federation.”

---

## [Decision Letter · Decision Letter 1]

27 Sep 2022

PONE-D-21-37203R1Immunogenicity and safety of a recombinant adenovirus type-5 COVID-19 vaccine in adults: data from a randomised, double-blind, placebo-controlled, single-dose, phase 3 trial in RussiaPLOS ONE

Dear Dr. Dzutseva,

Thank you for submitting your manuscript to PLOS ONE. After careful consideration, we feel that it has merit but does not fully meet PLOS ONE’s publication criteria as it currently stands. Therefore, we invite you to submit a revised version of the manuscript that addresses the points raised during the review process.

Minor revisions in various places of the manuscript are warranted, as suggested by the reviewer(s).

We look forward to receiving your revised manuscript.

Kind regards,

Nagendra R Hegde, B.V.Sc., M.V.Sc., Ph.D.

Academic Editor

PLOS ONE

Journal Requirements:

Reviewers' comments:

Reviewer's Responses to Questions

**Comments to the Author**

1. If the authors have adequately addressed your comments raised in a previous round of review and you feel that this manuscript is now acceptable for publication, you may indicate that here to bypass the “Comments to the Author” section, enter your conflict of interest statement in the “Confidential to Editor” section, and submit your "Accept" recommendation.

Reviewer #1: All comments have been addressed

Reviewer #3: (No Response)

2. Is the manuscript technically sound, and do the data support the conclusions?

Reviewer #1: Yes

Reviewer #3: Yes

3. Has the statistical analysis been performed appropriately and rigorously? 

Reviewer #1: Yes

Reviewer #3: Yes

4. Have the authors made all data underlying the findings in their manuscript fully available?

Reviewer #1: Yes

Reviewer #3: Yes

5. Is the manuscript presented in an intelligible fashion and written in standard English?

Reviewer #1: Yes

Reviewer #3: Yes

6. Review Comments to the Author

Reviewer #1: I endorse the publication of the manuscript "Immunogenicity and safety of a recombinant adenovirus type-5 COVID-19 vaccine in adults: data from a randomised, double-blind, placebo-controlled, single-dose, phase 3 trial in Russia" by Dzutseva et al.

Reviewer #3: The authors have addressed the majority of my earlier concerns. The revision is much easier to follow. There remain a few minor points:

1. Line 354: the seroconversion rate and 95% CI for NAbs is not correct (what is written is for RBD).

2. Lines 352-356: This sentence as worded is a little confusing, because the seroconversion rates are all from day 28 but the sentence talks about both day 14 and day 28.

3. Figure 3C (and related text and figure caption): The text says the comparison between the Ad4-nCoV group and placebo for NAbs was significant at 6 months (p<0.001), but the figure has a single * and the caption says * indicates p<0.05. Authors should either correct the figure or modify the sentence to match the results.

4. line 377: "most participants low" should be "most participants had low"

5. lines 379-380: authors added "Regardless of treatment group" to the start of the sentence, but the numbers seem to refer to the Ad5-nCoV group. Authors should clarify.

6. line 386: "based their" should be "based on their"

7. line 443: authors report a correlation of -0.05 at 6 months, but the dot in Figure 6 appears to be above the dashed line at 0. Is the correlation negative?

8. lines 444-447: I believe the authors want to refer to the magnitude of the correlation decreasing. Mathematically a negative number getting closer to 0 is actually increasing (but the magnitude would be the absolute value, so getting closer to 0 would be a decrease). Also, authors mention correlations decreasing from baseline to Month 6 for both S protein and nAbs, but there is not a correlation at baseline for nAbs in Figure 6.

9. line 552. I believe "As listed in S5 Table" should be "As listed in S6 table"

10. lines 555-556: the percentages do not match those presented in S6 Table

11. line 559: it is confusing to see 152/365 and 38/75. I realize authors added a note under Tables 2 and 3 to clarify, but in the text, wouldn't it be easier (and less confusing) to report just the number of participants - readers can see the number of AEs in the table.

7. PLOS authors have the option to publish the peer review history of their article (what does this mean?). If published, this will include your full peer review and any attached files.

Reviewer #1: No

Reviewer #3: No

---

## [Author Response · Author response to Decision Letter 1]

13 Oct 2022

Reviewer 3

Comment 1: Line 354: the seroconversion rate and 95% CI for NAbs is not correct (what is written is for RBD).

Authors’ Response: Thank you for bringing this to our attention. We have corrected the text based on the information presented in Table S1. The updated text is located in Line 357 in the revised manuscript as well as part of the response to Comment 2 as presented below.

Comment 2: Lines 352-356: This sentence as worded is a little confusing, because the seroconversion rates are all from day 28 but the sentence talks about both day 14 and day 28.

Authors’ Response: We acknowledge that this initial phrasing was not clear. We have revised the sentence to emphasize that the data presented is from the seroconversion rates for Day 28. The revised text in the Results section is as follows (pg. 19, lines 352–357):

“For those vaccinated with Ad5-nCoV vs. the placebo, seroconversion rates for S protein, RBD, and NAbs were significantly elevated by Day 14 (p<0.001; Fig. 3A–3C and S1 Table), and the seroconversion rates peaked at Day 28 for S protein (90.6% [95% CI: 87.2, 93.4] vs. 6.72% [95% CI: 2.95, 12.8]), RBD (78.5% [95% CI: 73.9, 82.6] vs. 5.88% [95% CI: 2.4, 11.7]), and NAbs (59.0% [95% CI: 53.3, 64.6] vs. 3.88% [95% CI: 1.07, 9.65]) (p<0.001).”

Comment 3: Figure 3C (and related text and figure caption): The text says the comparison between the Ad4-nCoV group and placebo for NAbs was significant at 6 months (p<0.001), but the figure has a single * and the caption says * indicates p<0.05. Authors should either correct the figure or modify the sentence to match the results.

Authors’ Response: We acknowledge the oversight on our part as the p-value of p<0.001 corresponded to only the seroconversion rate for S protein after 6 months when comparing those in the Ad5-nCoV vs the Placebo group. We have clarified the text to clearly attribute the p-value with the corresponding data in the figure. The revised text in the Results section is as follows (pg. 19, lines 357–359):

“Seroconversion rates for S protein and NAbs in the Ad5-nCoV group were elevated after 6 months compared with the Placebo group (p<0.001 and p<0.05, respectively; Fig. 3A and 3C, and S1 Table), indicating that some protection remained at the later timepoint.”

Comment 4: line 377: "most participants low" should be "most participants had low"

Authors’ Response: We have revised the text as per your suggestion.

Comment 5: lines 379-380: authors added "Regardless of treatment group" to the start of the sentence, but the numbers seem to refer to the Ad5-nCoV group. Authors should clarify.

Authors’ Response: We apologize for the confusion. We have revised the text for clarity. The revised text in the Results section is as follows (pg. 20, lines 379–381):

“Of those in the Ad5-nCoV group, GMTs of anti-Ad5 participants with low pre-existing anti-Ad5 antibodies increased from 11.3 at baseline (n=317) to 36.5 at Day 28 (n=353), which then increased further to 48.3 (n=355) at Month 6.”

Comment 6: line 386: "based their" should be "based on their"

Authors’ Response: We have revised the text as per your suggestion.

Comment 7: line 443: authors report a correlation of -0.05 at 6 months, but the dot in Figure 6 appears to be above the dashed line at 0. Is the correlation negative?

Authors’ Response: We originally intended to use a “+” for this data point. This has been corrected in the revised text (pg. 24, line 444).

Comment 8: lines 444-447: I believe the authors want to refer to the magnitude of the correlation decreasing. Mathematically a negative number getting closer to 0 is actually increasing (but the magnitude would be the absolute value, so getting closer to 0 would be a decrease). Also, authors mention correlations decreasing from baseline to Month 6 for both S protein and nAbs, but there is not a correlation at baseline for nAbs in Figure 6.

Authors’ Response: We have revised the text for clarity to emphasize the correlation magnitudes. We also specify that the timeframe as the GMTs of NAbs were recorded between Day 14 and Month 6. The revised text in the Results section is as follows (pg. 23, lines 444–448):

“The correlation magnitudes of GMTs to S protein and neutralising SARS-CoV-2 antibodies decreased from Day 14 to Month 6 (Fig. 6). These changes in GMT correlations indicate that the relationship between levels of baseline Ad5 antibodies to the COVID-19 humoral immune response diminishes over time following vaccination.”

Comment 9: line 552. I believe "As listed in S5 Table" should be "As listed in S6 table"

Authors’ Response: This has been corrected as S6 Table specifically lists the injection reactions by severity.

Comment 10: lines 555-556: the percentages do not match those presented in S6 Table

Authors’ Response: The total percentages were reported instead of the percentages of those specifically in the Ad5-nCoV group. We have revised the text accordingly. The revised text in the Results section is as follows (pg. 28, lines 554–557): 

“While moderate reactions rarely occurred in those that received the Ad5-nCoV vaccine, 15 participants (4%) in the Ad5-nCoV group experienced the following local post-vaccination reactions of moderate severity at the injection site: erythema (8 [2.2%]), pain (6 [1.6%]), swelling (5 [1.3%]), and induration (1 [0.3%]).”

Comment 11: line 559: it is confusing to see 152/365 and 38/75. I realize authors added a note under Tables 2 and 3 to clarify, but in the text, wouldn't it be easier (and less confusing) to report just the number of participants - readers can see the number of AEs in the table.

Authors’ Response: Thank you for this suggestion. We wanted to be as transparent as possible by reporting the numbers in the same format; however, this has created additional confusion. As per your suggestion we have reported only the number of participants. The revised text in the Results section is as follows (pg. 29, lines 560–562):

“As listed in Table 3, a total of 190 participants (38.3%) reported 440 AEs during the 6 months after vaccination, of which 152 (40.9%) participants from the Ad5-nCoV group reported 365 AEs and 38 (3.6%) participants from the Placebo group reported 75 AEs.”

---

## [Decision Letter · Decision Letter 2]

25 Nov 2022

Immunogenicity and safety of a recombinant adenovirus type-5 COVID-19 vaccine in adults: data from a randomised, double-blind, placebo-controlled, single-dose, phase 3 trial in Russia

PONE-D-21-37203R2

Dear Dr. Dzutseva,

We’re pleased to inform you that your manuscript has been judged scientifically suitable for publication and will be formally accepted for publication once it meets all outstanding technical requirements.

Kind regards,

Nagendra R Hegde, B.V.Sc., M.V.Sc., Ph.D.

Academic Editor

PLOS ONE

Additional Editor Comments (optional):

Reviewers' comments:

Reviewer's Responses to Questions

**Comments to the Author**

1. If the authors have adequately addressed your comments raised in a previous round of review and you feel that this manuscript is now acceptable for publication, you may indicate that here to bypass the “Comments to the Author” section, enter your conflict of interest statement in the “Confidential to Editor” section, and submit your "Accept" recommendation.

Reviewer #3: All comments have been addressed

2. Is the manuscript technically sound, and do the data support the conclusions?

Reviewer #3: (No Response)

3. Has the statistical analysis been performed appropriately and rigorously? 

Reviewer #3: (No Response)

4. Have the authors made all data underlying the findings in their manuscript fully available?

Reviewer #3: (No Response)

5. Is the manuscript presented in an intelligible fashion and written in standard English?

Reviewer #3: (No Response)

6. Review Comments to the Author

Reviewer #3: (No Response)

7. PLOS authors have the option to publish the peer review history of their article (what does this mean?). If published, this will include your full peer review and any attached files.

Reviewer #3: No

---

## [Editor Report · Acceptance letter]

28 Feb 2023

PONE-D-21-37203R2 

Immunogenicity and safety of a recombinant adenovirus type-5 COVID‑19 vaccine in adults: data from a randomised, double-blind, placebo-controlled, single-dose, phase 3 trial in Russia 

Dear Dr. Dzutseva:

I'm pleased to inform you that your manuscript has been deemed suitable for publication in PLOS ONE. Congratulations! Your manuscript is now with our production department. 

Kind regards, 

on behalf of

Dr. Nagendra R Hegde 

Academic Editor

PLOS ONE